# Deep Graph Matching Consensus

**Matthias Fey**[1,3] **Jan E. Lenssen**[1,4] **Christopher Morris**[1] **Jonathan Masci**[2] **Nils M. Kriege**[1]

[1]TU Dortmund University
Dortmund, Germany

[2]NNAISENSE
Lugano, Switzerland

## Abstract

This work presents a two-stage neural architecture for learning and refining structural correspondences between graphs. First, we use localized node embeddings computed by a graph neural network to obtain an initial ranking of soft correspondences between nodes. Secondly, we employ synchronous message passing networks to iteratively re-rank the soft correspondences to reach a matching consensus in local neighborhoods between graphs. We show, theoretically and empirically, that our message passing scheme computes a well-founded measure of consensus for corresponding neighborhoods, which is then used to guide the iterative re-ranking process. Our purely local and sparsity-aware architecture scales well to large, real-world inputs while still being able to recover global correspondences consistently. We demonstrate the practical effectiveness of our method on real-world tasks from the fields of computer vision and entity alignment between knowledge graphs, on which we improve upon the current state-of-the-art. Our source code is available under `https://github.com/rusty1s/deep-graph-matching-consensus`.

## 1 Introduction

Graph matching refers to the problem of establishing meaningful *structural correspondences* of nodes between two or more graphs by taking both node similarities and pairwise edge similarities into account (Wang et al., 2019b). Since graphs are natural representations for encoding relational data, the problem of graph matching lies at the heart of many real-world applications. For example, comparing molecules in cheminformatics (Kriege et al., 2019b), matching protein networks in bioinformatics (Sharan & Ideker, 2006; Singh et al., 2008), linking user accounts in social network analysis (Zhang & Philip, 2015), and tracking objects, matching 2D/3D shapes or recognizing actions in computer vision (Vento & Foggia, 2012) can be formulated as a graph matching problem.

The problem of graph matching has been heavily investigated in theory (Grohe et al., 2018) and practice (Conte et al., 2004), usually by relating it to domain-agnostic distances such as the *graph edit distance* (Stauffer et al., 2017) and the *maximum common subgraph* problem (Bunke & Shearer, 1998), or by formulating it as a *quadratic assignment problem* (Yan et al., 2016). Since all three approaches are NP-hard, solving them to optimality may not be tractable for large-scale, real-world instances. Moreover, these purely combinatorial approaches do not adapt to the given data distribution and often do not consider continuous node embeddings which can provide crucial information about node semantics.

Recently, various neural architectures have been proposed to tackle the task of graph matching (Zanfir & Sminchisescu, 2018; Wang et al., 2019b; Zhang & Lee, 2019; Xu et al., 2019d;b; Derr et al., 2019; Zhang et al., 2019a; Heimann et al., 2018) or graph similarity (Bai et al., 2018; 2019; Li et al., 2019) in a data-dependent fashion. However, these approaches are either only capable of computing similarity scores between whole graphs (Bai et al., 2018; 2019; Li et al., 2019), rely on an inefficient global matching procedure (Zanfir & Sminchisescu, 2018; Wang et al., 2019b; Xu et al., 2019d; Li et al., 2019), or do not generalize to unseen graphs (Xu et al., 2019b; Derr et al., 2019; Zhang et al., 2019a). Moreover, they might be prone to match neighborhoods between graphs

---
[3]Correspondence to `matthias.fey@udo.edu`
[4]Work done during an internship at NNAISENSE

inconsistently by only taking localized embeddings into account (Zanfir & Sminchisescu, 2018; Wang et al., 2019b; Zhang & Lee, 2019; Xu et al., 2019d; Derr et al., 2019; Heimann et al., 2018).

Here, we propose a fully-differentiable graph matching procedure which aims to reach a data-driven *neighborhood consensus* between matched node pairs without the need to solve any optimization problem during inference. In addition, our approach is *purely local*, *i.e.*, it operates on fixed-size neighborhoods around nodes, and is *sparsity-aware*, *i.e.*, it takes the sparsity of the underlying structures into account. Hence, our approach scales well to large input domains, and can be trained in an end-to-end fashion to adapt to a given data distribution. Finally, our approach improves upon the state-of-the-art on several real-world applications from the fields of computer vision and entity alignment on knowledge graphs.

## 2 PROBLEM DEFINITION

A *graph* $\mathcal{G} = (\mathcal{V}, \boldsymbol{A}, \boldsymbol{X}, \boldsymbol{E})$ consists of a finite set of *nodes* $\mathcal{V} = \{1, 2, \ldots\}$, an *adjacency matrix* $\boldsymbol{A} \in \{0, 1\}^{|\mathcal{V}| \times |\mathcal{V}|}$, a *node feature* matrix $\boldsymbol{X} \in \mathbb{R}^{|\mathcal{V}| \times \cdot}$, and an optional (sparse) *edge feature* matrix $\boldsymbol{E} \in \mathbb{R}^{|\mathcal{V}| \times |\mathcal{V}| \times \cdot}$. For a subset of nodes $\mathcal{S} \subseteq \mathcal{V}$, $\mathcal{G}[\mathcal{S}] = (\mathcal{S}, \boldsymbol{A}_{\mathcal{S}, \mathcal{S}}, \boldsymbol{X}_{\mathcal{S},:}, \boldsymbol{E}_{\mathcal{S}, \mathcal{S},:})$ denotes the *subgraph* of $\mathcal{G}$ induced by $\mathcal{S}$. We refer to $\mathcal{N}_T(i) = \{j \in \mathcal{V} \colon d(i, j) \leq T\}$ as the *$T$-hop neighborhood* around node $i \in \mathcal{V}$, where $d \colon \mathcal{V} \times \mathcal{V} \to \mathbb{N}$ denotes the shortest-path distance in $\mathcal{G}$. A *node coloring* is a function $\mathcal{V} \to \Sigma$ with arbitrary codomain $\Sigma$.

The problem of *graph matching* refers to establishing node correspondences between two graphs. Formally, we are given two graphs, a *source graph* $\mathcal{G}_s = (\mathcal{V}_s, \boldsymbol{A}_s, \boldsymbol{X}_s, \boldsymbol{E}_s)$ and a *target graph* $\mathcal{G}_t = (\mathcal{V}_t, \boldsymbol{A}_t, \boldsymbol{X}_t, \boldsymbol{E}_t)$, w.l.o.g. $|\mathcal{V}_s| \leq |\mathcal{V}_t|$, and are interested in finding a *correspondence matrix* $\boldsymbol{S} \in \{0, 1\}^{|\mathcal{V}_s| \times |\mathcal{V}_t|}$ which minimizes an objective subject to the one-to-one mapping constraints $\sum_{j \in \mathcal{V}_t} S_{i,j} = 1 \ \forall i \in \mathcal{V}_s$ and $\sum_{i \in \mathcal{V}_s} S_{i,j} \leq 1 \ \forall j \in \mathcal{V}_t$. As a result, $\boldsymbol{S}$ infers an injective mapping $\pi \colon \mathcal{V}_s \to \mathcal{V}_t$ which maps each node in $\mathcal{G}_s$ to a node in $\mathcal{G}_t$.

Typically, graph matching is formulated as an edge-preserving, quadratic assignment problem (Anstreicher, 2003; Gold & Rangarajan, 1996; Caetano et al., 2009; Cho et al., 2013), *i.e.*,

$$\operatorname*{argmax}_{\boldsymbol{S}} \sum_{\substack{i, i' \in \mathcal{V}_s \\ j, j' \in \mathcal{V}_t}} A_{i,i'}^{(s)} A_{j,j'}^{(t)} S_{i,j} S_{i',j'} \tag{1}$$

subject to the one-to-one mapping constraints mentioned above. This formulation is based on the intuition of finding correspondences based on *neighborhood consensus* (Rocco et al., 2018), which shall prevent adjacent nodes in the source graph from being mapped to different regions in the target graph. Formally, a neighborhood consensus is reached if for all node pairs $(i, j) \in \mathcal{V}_s \times \mathcal{V}_t$ with $S_{i,j} = 1$, it holds that for every node $i' \in \mathcal{N}_1(i)$ there exists a node $j' \in \mathcal{N}_1(j)$ such that $S_{i',j'} = 1$.

In this work, we consider the problem of supervised and semi-supervised matching of graphs while employing the intuition of neighborhood consensus as an inductive bias into our model. In the supervised setting, we are given pair-wise ground-truth correspondences for a set of graphs and want our model to generalize to unseen graph pairs. In the semi-supervised setting, source and target graphs are fixed, and ground-truth correspondences are only given for a small subset of nodes. However, we are allowed to make use of the complete graph structures.

## 3 METHODOLOGY

In the following, we describe our proposed end-to-end, deep graph matching architecture in detail. See Figure 1 for a high-level illustration. The method consists of two stages: a *local feature matching procedure* followed by an *iterative refinement strategy* using synchronous message passing networks. The aim of the feature matching step, see Section 3.1, is to compute initial correspondence scores based on the similarity of local node embeddings. The second step is an iterative refinement strategy, see Sections 3.2 and 3.3, which aims to reach neighborhood consensus for correspondences using a differentiable validator for graph isomorphism. Finally, in Section 3.4, we show how to scale our method to large, real-world inputs.

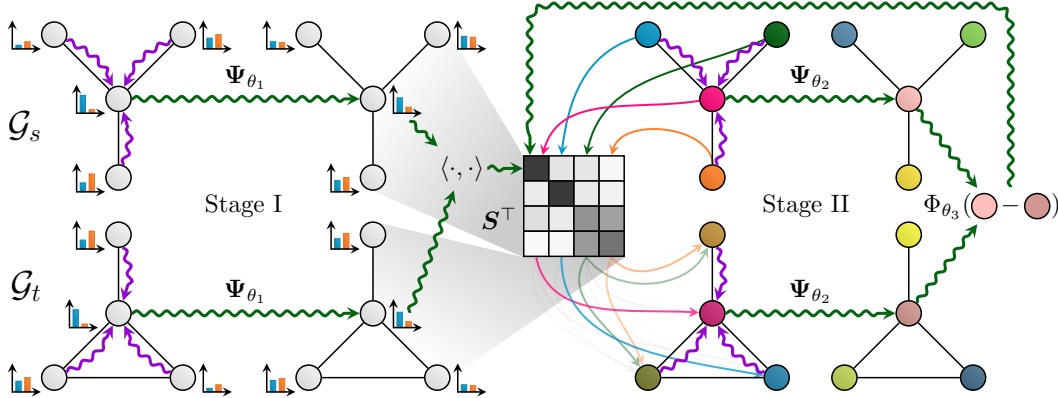

Figure 1: High-level illustration of our two-stage neighborhood consensus architecture. Node features are first locally matched based on a graph neural network $\mathbf{\Psi}_{\theta_1}$, before their correspondence scores get iteratively refined based on neighborhood consensus. Here, an injective node coloring of $\mathcal{G}_s$ is transferred to $\mathcal{G}_t$ via $\mathbf{S}$, and distributed by $\mathbf{\Psi}_{\theta_2}$ on both graphs. Updates on $\mathbf{S}$ are performed by a neural network $\Phi_{\theta_3}$ based on pair-wise color differences.

### 3.1 LOCAL FEATURE MATCHING

We model our *local feature matching procedure* in close analogy to related approaches (Bai et al., 2018; 2019; Wang et al., 2019b; Zhang & Lee, 2019; Wang & Solomon, 2019) by computing similarities between nodes in the source graph $\mathcal{G}_s$ and the target graph $\mathcal{G}_t$ based on node embeddings. That is, given latent node embeddings $\mathbf{H}_s = \mathbf{\Psi}_{\theta_1}(\mathbf{X}_s, \mathbf{A}_s, \mathbf{E}_s) \in \mathbb{R}^{|\mathcal{V}_s| \times \cdot}$ and $\mathbf{H}_t = \mathbf{\Psi}_{\theta_1}(\mathbf{X}_t, \mathbf{A}_t, \mathbf{E}_t) \in \mathbb{R}^{|\mathcal{V}_t| \times \cdot}$ computed by a shared neural network $\mathbf{\Psi}_{\theta_1}$ for source graph $\mathcal{G}_s$ and target graph $\mathcal{G}_t$, respectively, we obtain initial *soft* correspondences as

$$\mathbf{S}^{(0)} = \mathrm{sinkhorn}(\hat{\mathbf{S}}^{(0)}) \in [0,1]^{|\mathcal{V}_s| \times |\mathcal{V}_t|} \quad \text{with} \quad \hat{\mathbf{S}}^{(0)} = \mathbf{H}_s \mathbf{H}_t^\top \in \mathbb{R}^{|\mathcal{V}_s| \times |\mathcal{V}_t|}.$$

Here, $\mathrm{sinkhorn}$ normalization is applied to obtain *rectangular doubly-stochastic* correspondence matrices that fulfill the constraints $\sum_{j \in \mathcal{V}_t} S_{i,j} = 1 \ \forall i \in \mathcal{V}_s$ and $\sum_{i \in \mathcal{V}_s} S_{i,j} \le 1 \ \forall j \in \mathcal{V}_t$ (Sinkhorn & Knopp, 1967; Adams & Zemel, 2011; Cour et al., 2006).

We interpret the $i$-th row vector $\mathbf{S}_{i,:}^{(0)} \in [0,1]^{|\mathcal{V}_t|}$ as a discrete distribution over potential correspondences in $\mathcal{G}_t$ for each node $i \in \mathcal{V}_s$. We train $\mathbf{\Psi}_{\theta_1}$ in a dicriminative, supervised fashion against ground truth correspondences $\pi_{\mathrm{gt}}(\cdot)$ by minimizing the negative log-likelihood of correct correspondence scores $\mathcal{L}^{(\mathrm{initial})} = -\sum_{i \in \mathcal{V}_s} \log(S_{i,\pi_{\mathrm{gt}}(i)}^{(0)})$.

We implement $\mathbf{\Psi}_{\theta_1}$ as a *Graph Neural Network* (GNN) to obtain localized, permutation equivariant vectorial node representations (Bronstein et al., 2017; Hamilton et al., 2017; Battaglia et al., 2018; Goyal & Ferrara, 2018). Formally, a GNN follows a *neural message passing scheme* (Gilmer et al., 2017) and updates its node features $\vec{h}_i^{(t-1)}$ in layer $t$ by aggregating localized information via

$$\vec{a}_i^{(t)} = \mathrm{AGGREGATE}^{(t)}\left( \left\{\!\!\left\{ \left(\vec{h}_j^{(t-1)}, \vec{e}_{j,i}\right) : j \in \mathcal{N}_1(i) \right\}\!\!\right\} \right), \quad \vec{h}_i^{(t)} = \mathrm{UPDATE}^{(t)}\left(\vec{h}_i^{(t-1)}, \vec{a}_i^{(t)}\right) \quad (2)$$

where $\vec{h}_i^{(0)} = \vec{x}_i \in \mathbf{X}$ and $\{\!\!\{\ldots\}\!\!\}$ denotes a multiset. The recent work in the fields of *geometric deep learning* and *relational representation learning* provides a large number of operators to choose from (Kipf & Welling, 2017; Gilmer et al., 2017; Veličković et al., 2018; Schlichtkrull et al., 2018; Xu et al., 2019c), which allows for precise control of the properties of extracted features.

### 3.2 SYNCHRONOUS MESSAGE PASSING FOR NEIGHBORHOOD CONSENSUS

Due to the purely local nature of the used node embeddings, our feature matching procedure is prone to finding false correspondences which are locally similar to the correct one. Formally, those cases pose a violation of the neighborhood consensus criteria employed in Equation (1). Since finding a

global optimum is NP-hard, we aim to detect violations of the criteria in local neighborhoods and resolve them in an iterative fashion.

We utilize graph neural networks to detect these violations in a neighborhood consensus step and iteratively refine correspondences $S^{(l)}$, $l \in \{0, \ldots, L\}$, starting from $S^{(0)}$. Key to the proposed algorithm is the following observation: The soft correspondence matrix $S \in [0,1]^{|\mathcal{V}_s| \times |\mathcal{V}_t|}$ is a map from the node function space $L(\mathcal{G}_s) = L(\mathbb{R}^{|\mathcal{V}_s|})$ to the node function space $L(\mathcal{G}_t) = L(\mathbb{R}^{|\mathcal{V}_t|})$. Therefore, we can use $S$ to pass node functions $\vec{x}_s \in L(\mathcal{G}_s)$, $\vec{x}_t \in L(\mathcal{G}_t)$ along the soft correspondences by

$$\vec{x}_t' = S^\top \vec{x}_s \quad \text{and} \quad \vec{x}_s' = S\vec{x}_t \tag{3}$$

to obtain functions $\vec{x}_t' \in L(\mathcal{G}_t)$, $\vec{x}_s' \in L(\mathcal{G}_s)$ in the other domain, respectively.

Then, our consensus method works as follows: Using $S^{(l)}$, we first map node indicator functions, given as an injective node coloring $\mathcal{V}_s \to \{0,1\}^{|\mathcal{V}_s|}$ in the form of an identity matrix $I_{|\mathcal{V}_s|}$, from $\mathcal{G}_s$ to $\mathcal{G}_t$. Then, we distribute this coloring in corresponding neighborhoods by performing synchronous message passing on both graphs via a shared graph neural network $\Psi_{\theta_2}$, i.e.,

$$O_s = \Psi_{\theta_2}(I_{|\mathcal{V}_s|}, A_s, E_s) \quad \text{and} \quad O_t = \Psi_{\theta_2}(S_{(l)}^\top I_{|\mathcal{V}_s|}, A_t, E_t). \tag{4}$$

We can compare the results of both GNNs to recover a vector $\vec{d}_{i,j} = \vec{o}_i^{(s)} - \vec{o}_j^{(t)}$ which measures the neighborhood consensus between node pairs $(i,j) \in \mathcal{V}_s \times \mathcal{V}_t$. This measure can be used to perform trainable updates of the correspondence scores

$$S_{i,j}^{(l+1)} = \text{sinkhorn}(\hat{S}^{(l+1)})_{i,j} \quad \text{with} \quad \hat{S}_{i,j}^{(l+1)} = \hat{S}_{i,j}^{(l)} + \Phi_{\theta_3}(\vec{d}_{j,i}) \tag{5}$$

based on an MLP $\Phi_{\theta_3}$. The process can be applied $L$ times to iteratively improve the consensus in neighborhoods. The final objective $\mathcal{L} = \mathcal{L}^{\text{(initial)}} + \mathcal{L}^{\text{(refined)}}$ with $\mathcal{L}^{\text{(refined)}} = -\sum_{i \in \mathcal{V}_s} \log(S_{i,\pi_{\text{gt}}(i)}^{(L)})$ combines both the feature matching error and neighborhood consensus error. This objective is fully-differentiable and can hence be optimized in an end-to-end fashion using stochastic gradient descent. Overall, the consensus stage distributes global node colorings to resolve ambiguities and false matchings made in the first stage of our architecture by only using purely local operators. Since an initial matching is needed to test for neighborhood consensus, this task cannot be fulfilled by $\Psi_{\theta_1}$ alone, which stresses the importance of our two-stage approach.

The following two theorems show that $\vec{d}_{i,j}$ is a good measure of how well local neighborhoods around $i$ and $j$ are matched by the soft correspondence between $\mathcal{G}_s$ and $\mathcal{G}_t$. The proofs can be found in Appendix B and C, respectively.

**Theorem 1.** *Let $\mathcal{G}_s$ and $\mathcal{G}_t$ be two isomorphic graphs and let $\Psi_{\theta_2}$ be a permutation equivariant GNN, i.e., $P^\top \Psi_{\theta_2}(X, A) = \Psi_{\theta_2}(P^\top X, P^\top A P)$ for any permutation matrix $P \in \{0,1\}^{|\mathcal{V}| \times |\mathcal{V}|}$. If $S \in \{0,1\}^{|\mathcal{V}_s| \times |\mathcal{V}_t|}$ encodes an isomorphism between $\mathcal{G}_s$ and $\mathcal{G}_t$, then $\vec{d}_{i,\pi(i)} = \vec{0}$ for all $i \in \mathcal{V}_s$.*

**Theorem 2.** *Let $\mathcal{G}_s$ and $\mathcal{G}_t$ be two graphs and let $\Psi_{\theta_2}$ be a permutation equivariant and $T$-layered GNN for which both $\text{AGGREGATE}^{(t)}$ and $\text{UPDATE}^{(t)}$ are injective for all $t \in \{1, \ldots, T\}$. If $\vec{d}_{i,j} = \vec{0}$, then the resulting submatrix $S_{\mathcal{N}_T(i), \mathcal{N}_T(j)} \in [0,1]^{|\mathcal{N}_T(i)| \times |\mathcal{N}_T(j)|}$ is a permutation matrix describing an isomorphism between the $T$-hop subgraph $\mathcal{G}_s[\mathcal{N}_T(i)]$ around $i \in \mathcal{V}_s$ and the $T$-hop subgraph $\mathcal{G}_t[\mathcal{N}_T(j)]$ around $j \in \mathcal{V}_t$. Moreover, if $\vec{d}_{i,\text{argmax } S_{i,:}} = \vec{0}$ for all $i \in \mathcal{V}_s$, then $S$ denotes a full isomorphism between $\mathcal{G}_s$ and $\mathcal{G}_t$.*

Hence, a GNN $\Psi_{\theta_2}$ that satisfies both criteria in Theorem 1 and 2 provides equal node embeddings $\vec{o}_i^{(s)}$ and $\vec{o}_j^{(t)}$ if and only if nodes in a local neighborhood are correctly matched to each other. A value $\vec{d}_{i,j} \neq \vec{0}$ indicates the existence of inconsistent matchings in the local neighborhoods around $i$ and $j$, and can hence be used to refine the correspondence score $\hat{S}_{i,j}$.

Note that both requirements, permutation equivariance and injectivity, are easily fulfilled: (1) All common graph neural network architectures following the message passing scheme of Equation (2) are equivariant due to the use of permutation invariant neighborhood aggregators. (2) Injectivity of graph neural networks is a heavily discussed topic in recent literature. It can be fulfilled by using a GNN that is as powerful as the Weisfeiler & Lehman (1968) (WL) heuristic in distinguishing graph structures, e.g., by using $\text{sum}$ aggregation in combination with $\text{MLP}$s on the multiset of neighboring node features, cf. (Xu et al., 2019c; Morris et al., 2019).

### 3.3 Relation to the Graduated Assignment Algorithm

Theoretically, we can relate our proposed approach to classical graph matching techniques that consider a doubly-stochastic relaxation of the problem defined in Equation (1), *cf.* (Lyzinski et al., 2016) and Appendix F for more details. A seminal work following this method is the *graduated assignment algorithm* (Gold & Rangarajan, 1996). By starting from an initial feasible solution $\boldsymbol{S}^{(0)}$, a new solution $\boldsymbol{S}^{(l+1)}$ is iteratively computed from $\boldsymbol{S}^{(l)}$ by approximately solving a linear assignment problem according to

$$\boldsymbol{S}^{(l+1)} \leftarrow \underset{\boldsymbol{S}}{\mathrm{softassign}} \sum_{i \in \mathcal{V}_s} \sum_{j \in \mathcal{V}_t} Q_{i,j} S_{i,j} \quad \text{with} \quad Q_{i,j} = 2 \sum_{i' \in \mathcal{V}_s} \sum_{j' \in \mathcal{V}_t} A_{i,i'}^{(s)} A_{j,j'}^{(t)} S_{i',j'}^{(l)} \tag{6}$$

where $\boldsymbol{Q}$ denotes the gradient of Equation (1) at $\boldsymbol{S}^{(l)}$.[1] The $\mathrm{softassign}$ operator is implemented by applying $\mathrm{sinkhorn}$ normalization on rescaled inputs, where the scaling factor grows in every iteration to increasingly encourage integer solutions. Our approach also resembles the approximation of the linear assignment problem via $\mathrm{sinkhorn}$ normalization.

Moreover, the gradient $\boldsymbol{Q}$ is closely related to our neighborhood consensus scheme for the particular simple, non-trainable GNN instantiation $\boldsymbol{\Psi}(\boldsymbol{X}, \boldsymbol{A}, \boldsymbol{E}) = \boldsymbol{A}\boldsymbol{X}$. Given $\boldsymbol{O}_s = \boldsymbol{A}_s \boldsymbol{I}_{|\mathcal{V}_s|} = \boldsymbol{A}_s$ and $\boldsymbol{O}_t = \boldsymbol{A}_t \boldsymbol{S}^\top \boldsymbol{I}_{|\mathcal{V}_s|} = \boldsymbol{A}_t \boldsymbol{S}^\top$, we obtain $\boldsymbol{Q} = 2 \boldsymbol{O}_s \boldsymbol{O}_t^\top$ by substitution. Instead of updating $\boldsymbol{S}^{(l)}$ based on the similarity between $\boldsymbol{O}_s$ and $\boldsymbol{O}_t$ obtained from a fixed-function GNN $\boldsymbol{\Psi}$, we choose to update correspondence scores via trainable neural networks $\boldsymbol{\Psi}_{\theta_2}$ and $\Phi_{\theta_3}$ based on the difference between $\boldsymbol{O}_s$ and $\boldsymbol{O}_t$. This allows us to interpret our model as a deep parameterized generalization of the graduated assignment algorithm. In addition, specifying node and edge attribute similarities in graph matching is often difficult and complicates its computation (Zhou & De la Torre, 2016; Zhang et al., 2019c), whereas our approach naturally supports continuous node and edge features via established GNN models. We experimentally verify the benefits of using trainable neural networks $\boldsymbol{\Psi}_{\theta_2}$ instead of $\boldsymbol{\Psi}(\boldsymbol{X}, \boldsymbol{A}, \boldsymbol{E}) = \boldsymbol{A}\boldsymbol{X}$ in Appendix D.

### 3.4 Scaling to Large Input

We apply a number of optimizations to our proposed algorithm to make it scale to large input domains. See Algorithm 1 in Appendix A for the final optimized algorithm.

**Sparse correspondences.** We propose to sparsify initial correspondences $\boldsymbol{S}^{(0)}$ by filtering out low score correspondences before neighborhood consensus takes place. That is, we sparsify $\boldsymbol{S}^{(0)}$ by computing top $k$ correspondences with the help of the KEOPS library (Charlier et al., 2019) without ever storing its dense version, reducing its required memory footprint from $\mathcal{O}(|\mathcal{V}_s||\mathcal{V}_t|)$ to $\mathcal{O}(k|\mathcal{V}_s|)$. In addition, the time complexity of the refinement phase is reduced from $\mathcal{O}(|\mathcal{V}_s||\mathcal{V}_t| + |\mathcal{E}_s| + |\mathcal{E}_t|)$ to $\mathcal{O}(k|\mathcal{V}_s| + |\mathcal{E}_s| + |\mathcal{E}_t|)$, where $|\mathcal{E}_s|$ and $|\mathcal{E}_t|$ denote the number of edges in $\mathcal{G}_s$ and $\mathcal{G}_t$, respectively. Note that sparsifying initial correspondences assumes that the feature matching procedure ranks the correct correspondence within the top $k$ elements for each node $i \in \mathcal{V}_s$. Hence, also optimizing the initial feature matching loss $\mathcal{L}^{(\text{initial})}$ is crucial, and can be further accelerated by training only against sparsified correspondences with ground-truth entries $\mathrm{top}_k(\boldsymbol{S}_{i,:}^{(0)}) \cup \{S_{i,\pi_{\text{gt}}(i)}^{(0)}\}$.

**Replacing node indicators functions.** Although applying $\boldsymbol{\Psi}_{\theta_2}$ on node indicator functions $\boldsymbol{I}_{|\mathcal{V}_s|}$ is computationally efficient, it requires a parameter complexity of $\mathcal{O}(|\mathcal{V}_s|)$. Hence, we propose to replace node indicator functions $\boldsymbol{I}_{|\mathcal{V}_s|}$ with randomly drawn node functions $\boldsymbol{R}_s^{(l)} \sim \mathcal{N}(0, 1)$, where $\boldsymbol{R}_s^{(l)} \in \mathbb{R}^{|\mathcal{V}_s| \times r}$ with $r \ll |\mathcal{V}_s|$, in iteration $l$. By sampling from a continuous distribution, node indicator functions are still guaranteed to be injective (DeGroot & Schervish, 2012). Note that Theorem 1 still holds because it does not impose any restrictions on the function space $L(\mathcal{G}_s)$. Theorem 2 does not necessarily hold anymore, but we expect our refinement strategy to resolve any ambiguities by re-sampling $\boldsymbol{R}_s^{(l)}$ in every iteration $l$. We verify this empirically in Section 4.1.

---

[1]For clarity of presentation, we closely follow the original formulation of the method for simple graphs but ignore the edge similarities and adapt the constant factor of the gradient according to our objective function.

**Softmax normalization.** The sinkhorn normalization fulfills the requirements of rectangular doubly-stochastic solutions. However, it may eventually push correspondences to inconsistent integer solutions very early on from which the neighborhood consensus method cannot effectively recover. Furthermore, it is inherently inefficient to compute and runs the risk of vanishing gradients $\partial \boldsymbol{S}^{(l)}/\partial \hat{\boldsymbol{S}}^{(l)}$ (Zhang et al., 2019b). Here, we propose to relax this constraint by only applying row-wise softmax normalization on $\hat{\boldsymbol{S}}^{(l)}$, and expect our supervised refinement procedure to naturally resolve violations of $\sum_{i \in \mathcal{V}_s} S_{i,j} \leq 1$ on its own by re-ranking false correspondences via neighborhood consensus. Experimentally, we show that row-wise normalization is sufficient for our algorithm to converge to the correct solution, *cf.* Section 4.1.

**Number of refinement iterations.** Instead of holding $L$ fixed, we propose to differ the number of refinement iterations $L^{(\text{train})}$ and $L^{(\text{test})}$, $L^{(\text{train})} \ll L^{(\text{test})}$, for training and testing, respectively. This does not only speed up training runtime, but it also encourages the refinement procedure to reach convergence with as few steps as necessary while we can run the refinement procedure until convergence during testing. We show empirically that decreasing $L^{(\text{train})}$ does not affect the convergence abilities of our neighborhood consensus procedure during testing, *cf.* Section 4.1.

## 4 EXPERIMENTS

We verify our method on three different tasks. We first show the benefits of our approach in an ablation study on synthetic graphs (Section 4.1), and apply it to the real-world tasks of supervised keypoint matching in natural images (Sections 4.2 and 4.3) and semi-supervised cross-lingual knowledge graph alignment (Section 4.4) afterwards. All dataset statistics can be found in Appendix H.

Our method is implemented in PYTORCH (Paszke et al., 2017) using the PYTORCH GEOMETRIC (Fey & Lenssen, 2019) and the KEOPS (Charlier et al., 2019) libraries. Our implementation can process sparse mini-batches with parallel GPU acceleration and minimal memory footprint in all algorithm steps. For all experiments, optimization is done via ADAM (Kingma & Ba, 2015) with a fixed learning rate of $10^{-3}$. We use similar architectures for $\boldsymbol{\Psi}_{\theta_1}$ and $\boldsymbol{\Psi}_{\theta_2}$ except that we omit dropout (Srivastava et al., 2014) in $\boldsymbol{\Psi}_{\theta_2}$. For all experiments, we report Hits@$k$ to evaluate and compare our model to previous lines of work, where Hits@$k$ measures the proportion of correctly matched entities ranked in the top $k$.

### 4.1 ABLATION STUDY ON SYNTHETIC GRAPHS

In our first experiment, we evaluate our method on synthetic graphs where we aim to learn a matching for pairs of graphs in a supervised fashion. Each pair of graphs consists of an undirected Erdős & Rényi (1959) graph $\mathcal{G}_s$ with $|\mathcal{V}_s| \in \{50, 100\}$ nodes and edge probability $p \in \{0.1, 0.2\}$, and a target graph $\mathcal{G}_t$ which is constructed from $\mathcal{G}_s$ by removing edges with probability $p_s$ without disconnecting any nodes (Heimann et al., 2018). Training and evaluation is done on $1\,000$ graphs each for different configurations $p_s \in \{0.0, 0.1, 0.2, 0.3, 0.4, 0.5\}$. In Appendix E, we perform additional experiments to also verify the robustness of our approach towards node addition or removal.

**Architecture and parameters.** We implement the graph neural network operators $\boldsymbol{\Psi}_{\theta_1}$ and $\boldsymbol{\Psi}_{\theta_2}$ by stacking three layers ($T = 3$) of the GIN operator (Xu et al., 2019c)

$$\vec{h}_i^{(t+1)} = \text{MLP}^{(t+1)}\left( \left(1 + \epsilon^{(t+1)}\right) \cdot \vec{h}_i^{(t)} + \sum_{j \to i} \vec{h}_j^{(t)} \right) \tag{7}$$

due to its expressiveness in distinguishing raw graph structures. The number of layers and hidden dimensionality of all MLPs is set to 2 and 32, respectively, and we apply ReLU activation (Glorot et al., 2011) and Batch normalization (Ioffe & Szegedy, 2015) after each of its layers. Input features are initialized with one-hot encodings of node degrees. We employ a *Jumping Knowledge* style concatenation $\vec{h}_i = \boldsymbol{W}[\vec{h}_i^{(1)}, \ldots, \vec{h}_i^{(T)}]$ (Xu et al., 2018) to compute final node representations $\vec{h}_i$. We train and test our procedure with $L^{(\text{train})} = 10$ and $L^{(\text{test})} = 20$ refinement iterations, respectively.

**Results.** Figures 2(a) and 2(b) show the matching accuracy Hits@1 for different choices of $|\mathcal{V}_s|$ and $p$. We observe that the purely local matching approach via softmax($\hat{\boldsymbol{S}}^{(0)}$) starts decreasing in

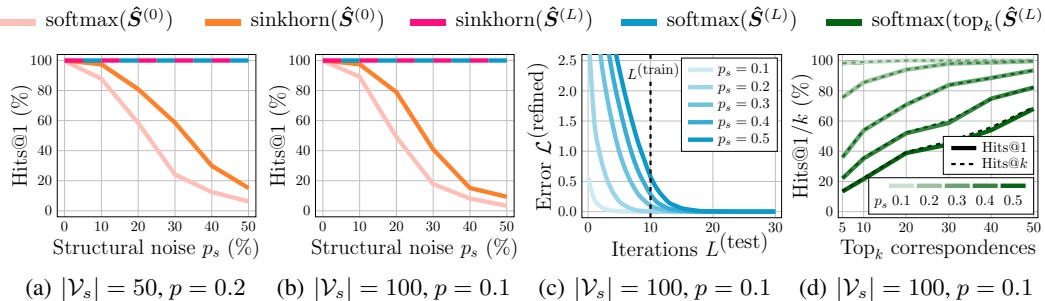

Figure 2: The performance of our method on synthetic data with structural noise.

performance with the structural noise $p_s$ increasing. This also holds when applying global sinkhorn normalization on $\hat{\boldsymbol{S}}^{(0)}$. However, our proposed two-stage architecture can recover *all* correspondences, independent of the applied structural noise $p_s$. This applies to both variants discussed in the previous sections, *i.e.*, our initial formulation $\mathrm{sinkhorn}(\hat{\boldsymbol{S}}^{(L)})$, and our optimized architecture using random node indicator sampling and row-wise normalization $\mathrm{softmax}(\hat{\boldsymbol{S}}^{(L)})$. This highlights the overall benefits of applying matching consensus and justifies the usage of the enhancements made towards scalability in Section 3.4.

In addition, Figure 2(c) visualizes the test error $\mathcal{L}^{(\mathrm{refined})}$ for varying number of iterations $L^{(\mathrm{test})}$. We observe that even when training to non-convergence, our procedure is still able to converge by increasing the number of iterations $L^{(\mathrm{test})}$ during testing.

Moreover, Figure 2(d) shows the performance of our refinement strategy when operating on sparsified top $k$ correspondences. In contrast to its dense version, it cannot match all nodes correctly due to the poor initial feature matching quality. However, it consistently converges to the perfect solution of Hits@1 $\approx$ Hits@$k$ in case the correct match is included in the initial top $k$ ranking of correspondences. Hence, with increasing $k$, we can recover most of the correct correspondences, making it an excellent option to scale our algorithm to large graphs, *cf.* Section 4.4.

## 4.2 SUPERVISED KEYPOINT MATCHING IN NATURAL IMAGES

We perform experiments on the PASCALVOC (Everingham et al., 2010) with Berkeley annotations (Bourdev & Malik, 2009) and WILLOW-OBJECTCLASS (Cho et al., 2013) datasets which contain sets of image categories with labeled keypoint locations. For PASCALVOC, we follow the experimental setups of Zanfir & Sminchisescu (2018) and Wang et al. (2019b) and use the training and test splits provided by Choy et al. (2016). We pre-filter the dataset to exclude difficult, occluded and truncated objects, and require examples to have at least one keypoint, resulting in 6 953 and 1 671 annotated images for training and testing, respectively. The PASCALVOC dataset contains instances of varying scale, pose and illumination, and the number of keypoints ranges from 1 to 19. In contrast, the WILLOW-OBJECTCLASS dataset contains at least 40 images with consistent orientations for each of its five categories, and each image consists of exactly 10 keypoints. Following the experimental setup of peer methods (Cho et al., 2013; Wang et al., 2019b), we pre-train our model on PASCALVOC and fine-tune it over 20 random splits with 20 per-class images used for training. We construct graphs via the Delaunay triangulation of keypoints. For fair comparison with Zanfir & Sminchisescu (2018) and Wang et al. (2019b), input features of keypoints are given by the concatenated output of `relu4_2` and `relu5_1` of a pre-trained VGG16 (Simonyan & Zisserman, 2014) on IMAGENET (Deng et al., 2009).

**Architecture and parameters.** We adopt SPLINECNN (Fey et al., 2018) as our graph neural network operator

$$\vec{h}_i^{(t+1)} = \sigma\left(\boldsymbol{W}^{(t+1)}\vec{h}_i^{(t)} + \sum_{j \to i} \boldsymbol{\Phi}_\theta^{(t+1)}(\vec{e}_{j,i}) \cdot \vec{h}_j^{(t)}\right) \tag{8}$$

whose trainable B-spline based kernel function $\boldsymbol{\Phi}_\theta(\cdot)$ is conditioned on edge features $\vec{e}_{j,i}$ between node-pairs. To align our results with the related work, we evaluate both isotropic and anisotropic

Table 1: Hits@1 (%) on the PASCALVOC dataset with Berkeley keypoint annotations.

| Method | | | Aero | Bike | Bird | Boat | Bottle | Bus | Car | Cat | Chair | Cow | Table | Dog | Horse | M-Bike | Person | Plant | Sheep | Sofa | Train | TV | Mean |
|---|---|---|---|---|---|---|---|---|---|---|---|---|---|---|---|---|---|---|---|---|---|---|---|
| GMN | | | 31.1 | 46.2 | 58.2 | 45.9 | 70.6 | 76.5 | 61.2 | 61.7 | 35.5 | 53.7 | 58.9 | 57.5 | 56.9 | 49.3 | 34.1 | 77.5 | 57.1 | 53.6 | 83.2 | 88.6 | 57.9 |
| PCA-GM | | | 40.9 | 55.0 | **65.8** | 47.9 | 76.9 | 77.9 | 63.5 | 67.4 | 33.7 | **66.5** | 63.6 | **61.3** | 58.9 | **62.8** | 44.9 | 77.5 | 67.4 | 57.5 | 86.7 | **90.9** | 63.8 |
| $\Psi_{\theta_1} = $ MLP isotropic | | $L=0$ | 34.7 | 42.6 | 41.5 | 50.4 | 50.3 | 72.2 | 60.1 | 59.4 | 24.6 | 38.1 | 86.2 | 47.7 | 56.3 | 37.6 | 35.4 | 58.0 | 45.8 | 74.8 | 64.1 | 75.3 | 52.8 |
| | | $L=10$ | 45.8 | 58.2 | 45.5 | 57.6 | 68.2 | 82.1 | 75.3 | 60.2 | 31.7 | 52.9 | **88.2** | 56.2 | 68.2 | 50.7 | 46.5 | 66.3 | 58.8 | 89.0 | 85.1 | 79.9 | 63.3 |
| | | $L=20$ | 45.3 | 57.1 | 54.9 | 54.7 | 71.7 | 82.6 | 75.3 | 65.9 | 31.6 | 50.8 | 86.1 | 56.9 | 67.1 | 53.1 | 49.2 | 77.3 | 59.2 | 91.7 | 82.0 | 84.2 | 64.8 |
| $\Psi_{\theta_1} = $ GNN isotropic | | $L=0$ | 44.3 | 62.0 | 48.4 | 53.9 | 73.3 | 80.4 | 72.2 | 64.2 | 30.3 | 52.7 | 79.4 | 56.6 | 62.3 | 56.2 | 47.5 | 74.0 | 59.8 | 79.9 | 81.9 | 83.0 | 63.1 |
| | | $L=10$ | 46.5 | 63.7 | 54.9 | 60.9 | 79.4 | **84.1** | 76.4 | 68.3 | **38.5** | 61.5 | 80.6 | 59.7 | 69.8 | 58.4 | 54.3 | 76.4 | 64.5 | **95.7** | **87.9** | 81.3 | 68.1 |
| | | $L=20$ | **50.1** | **65.4** | 55.7 | **65.3** | **80.0** | 83.5 | **78.3** | **69.7** | 34.7 | 60.7 | 70.4 | 59.9 | **70.0** | 62.2 | 56.1 | **80.2** | **70.3** | 88.8 | 81.1 | 84.3 | **68.3** |
| $\Psi_{\theta_1} = $ MLP anisotropic | | $L=0$ | 34.3 | 45.9 | 37.3 | 47.7 | 53.3 | 75.2 | 64.5 | 61.7 | 27.7 | 40.5 | 85.9 | 46.6 | 50.2 | 39.0 | 37.3 | 58.0 | 49.2 | 82.9 | 65.0 | 74.2 | 53.8 |
| | | $L=10$ | 44.6 | 51.2 | 50.7 | 58.5 | 72.3 | 83.3 | 76.6 | 65.6 | 31.0 | 57.5 | 91.7 | 55.4 | 69.5 | 56.2 | 47.5 | 85.1 | 57.9 | 92.3 | 86.7 | 85.9 | 66.0 |
| | | $L=20$ | **48.7** | 57.2 | 47.0 | 65.3 | 73.9 | 87.6 | 76.7 | 70.0 | 30.0 | 55.5 | **92.8** | 59.5 | 67.9 | 56.9 | 48.7 | 87.2 | 58.3 | 94.9 | 87.9 | 86.0 | 67.6 |
| $\Psi_{\theta_1} = $ GNN anisotropic | | $L=0$ | 42.1 | 57.5 | 49.6 | 59.4 | 83.8 | 84.0 | 78.4 | 67.5 | 37.3 | 60.4 | 85.0 | 58.0 | 66.0 | 54.1 | 52.6 | 93.9 | 60.2 | 85.6 | 87.8 | 82.5 | 67.3 |
| | | $L=10$ | 45.5 | **67.6** | 56.5 | 66.8 | **86.9** | 85.2 | 84.2 | **73.0** | **43.6** | 66.0 | 92.3 | **64.0** | **79.8** | 56.6 | 56.1 | 95.4 | 64.4 | **95.0** | 91.3 | 86.3 | 72.8 |
| | | $L=20$ | 47.0 | 65.7 | 56.8 | **67.6** | **86.9** | **87.7** | **85.3** | 72.6 | 42.9 | **69.1** | 84.5 | 63.8 | 78.1 | 55.6 | **58.4** | **98.0** | **68.4** | 92.2 | **94.5** | 85.5 | **73.0** |

Table 2: Hits@1 (%) with standard deviations on the WILLOW-OBJECTCLASS dataset.

| Method | | | Face | Motorbike | Car | Duck | Winebottle |
|---|---|---|---|---|---|---|---|
| GMN (Zanfir & Sminchisescu, 2018) | | | 99.3 | 71.4 | 74.3 | 82.8 | 76.7 |
| PCA-GM (Wang et al., 2019b) | | | **100.0** | 76.7 | 84.0 | **93.5** | 96.9 |
| $\Psi_{\theta_1} = $ MLP | isotropic | $L=0$ | 98.07 ± 0.79 | 48.97 ± 4.62 | 65.30 ± 3.16 | 66.02 ± 2.51 | 77.72 ± 3.32 |
| | | $L=10$ | **100.00 ± 0.00** | 67.28 ± 4.93 | 85.07 ± 3.93 | 83.10 ± 3.61 | 92.30 ± 2.11 |
| | | $L=20$ | **100.00 ± 0.00** | 68.57 ± 3.94 | 82.75 ± 5.77 | 84.18 ± 4.15 | 90.36 ± 2.42 |
| $\Psi_{\theta_1} = $ GNN | isotropic | $L=0$ | 99.62 ± 0.28 | 73.47 ± 3.32 | 77.47 ± 4.92 | 77.10 ± 3.25 | 88.04 ± 1.38 |
| | | $L=10$ | **100.00 ± 0.00** | **92.05 ± 3.49** | 90.05 ± 5.10 | 88.98 ± 2.75 | **97.14 ± 1.41** |
| | | $L=20$ | **100.00 ± 0.00** | **92.05 ± 3.24** | **90.28 ± 4.67** | 88.97 ± 3.49 | **97.14 ± 1.83** |
| $\Psi_{\theta_1} = $ MLP | anisotropic | $L=0$ | 98.47 ± 0.61 | 49.28 ± 4.31 | 64.95 ± 3.52 | 66.17 ± 4.08 | 78.08 ± 2.61 |
| | | $L=10$ | **100.00 ± 0.00** | 76.28 ± 4.77 | 86.70 ± 3.25 | 83.22 ± 3.52 | 93.65 ± 1.64 |
| | | $L=20$ | **100.00 ± 0.00** | 76.57 ± 5.28 | 89.00 ± 3.88 | 84.78 ± 2.73 | 95.29 ± 2.22 |
| $\Psi_{\theta_1} = $ GNN | anisotropic | $L=0$ | 99.96 ± 0.06 | 91.90 ± 2.30 | 91.28 ± 4.89 | 86.58 ± 2.99 | 98.25 ± 0.71 |
| | | $L=10$ | **100.00 ± 0.00** | 98.80 ± 1.58 | **96.53 ± 1.55** | 93.22 ± 3.77 | **99.87 ± 0.31** |
| | | $L=20$ | **100.00 ± 0.00** | **99.40 ± 0.80** | 95.53 ± 2.93 | 93.00 ± 2.71 | 99.39 ± 0.70 |

edge features which are given as normalized relative distances and 2D Cartesian coordinates, respectively. For SPLINECNN, we use a kernel size of 5 in each dimension, a hidden dimensionality of 256, and apply ReLU as our non-linearity function $\sigma$. Our network architecture consists of two convolutional layers ($T = 2$), followed by dropout with probability 0.5, and a final linear layer. During training, we form pairs between any two training examples of the same category, and evaluate our model by sampling a fixed number of test graph pairs belonging to the same category.

**Results.** We follow the experimental setup of Wang et al. (2019b) and train our models using negative log-likelihood due to its superior performance in contrast to the *displacement loss* used in Zanfir & Sminchisescu (2018). We evaluate our complete architecture using isotropic and anisotropic GNNs for $L \in \{0, 10, 20\}$, and include ablation results obtained from using $\Psi_{\theta_1} = $ MLP for the local node matching procedure. Results of Hits@1 are shown in Table 1 and 2 for PASCALVOC and WILLOW-OBJECTCLASS, respectively. We visualize qualitative results of our method in Appendix I.

We observe that our refinement strategy is able to significantly outperform competing methods as well as our non-refined baselines. On the WILLOW-OBJECTCLASS dataset, our refinement stage at least reduces the error of the initial model ($L = 0$) by half across all categories. The benefits of the second stage are even more crucial when starting from a weaker initial feature matching baseline ($\Psi_{\theta_1} = $ MLP), with overall improvements of up to 14 percentage points on PASCALVOC. However, good initial matchings do help our consensus stage to improve its performance further, as indicated by the usage of task-specific isotropic or anisotropic GNNs for $\Psi_{\theta_1}$.

Table 3: Hits@1 (%) on the PASCALPF dataset using a synthetic training setup.

| Method | | Aero | Bike | Bird | Boat | Bottle | Bus | Car | Cat | Chair | Cow | Table | Dog | Horse | M-Bike | Person | Plant | Sheep | Sofa | Train | TV | Mean |
|---|---|---|---|---|---|---|---|---|---|---|---|---|---|---|---|---|---|---|---|---|---|---|
| (Zhang & Lee, 2019) | | 76.1 | 89.8 | 93.4 | 96.4 | 96.2 | 97.1 | 94.6 | 82.8 | 89.3 | **96.7** | 89.7 | 79.5 | 82.6 | 83.5 | 72.8 | 76.7 | 77.1 | 97.3 | 98.2 | **99.5** | 88.5 |
| | $L = 0$ | 69.2 | 87.7 | 77.3 | 90.4 | 98.7 | 98.3 | 92.5 | 91.6 | 94.7 | 79.4 | 95.8 | 90.1 | 80.0 | 79.5 | 72.5 | 98.0 | 76.5 | 89.6 | 93.4 | 97.8 | 87.6 |
| **Ours** | $L = 10$ | **81.3** | **92.2** | 94.2 | 98.8 | **99.3** | 99.1 | 98.6 | **98.2** | **99.6** | 94.1 | **100.0** | **99.4** | **86.6** | **86.6** | **88.7** | **100.0** | **100.0** | **100.0** | **100.0** | 99.3 | **95.8** |
| | $L = 20$ | 81.1 | 92.0 | **94.7** | **100.0** | **99.3** | **99.3** | **98.9** | 97.3 | 99.4 | 93.4 | **100.0** | 99.1 | 86.3 | 86.2 | **87.7** | **100.0** | **100.0** | **100.0** | **100.0** | 99.3 | 95.7 |

## 4.3 SUPERVISED GEOMETRIC KEYPOINT MATCHING

We also verify our approach by tackling the *geometric feature matching problem*, where we only make use of point coordinates and no additional visual features are available. Here, we follow the experimental training setup of Zhang & Lee (2019), and test the generalization capabilities of our model on the PASCALPF dataset (Ham et al., 2016). For training, we generate a synthetic set of graph pairs: We first randomly sample 30–60 source points uniformly from $[-1, 1]^2$, and add Gaussian noise from $\mathcal{N}(0, 0.05^2)$ to these points to obtain the target points. Furthermore, we add 0–20 outliers from $[-1.5, 1.5]^2$ to each point cloud. Finally, we construct graphs by connecting each node with its $k$-nearest neighbors ($k = 8$). We train our unmodified anisotropic keypoint architecture from Section 4.2 with input $\vec{x}_i = \vec{1} \in \mathbb{R}^1 \, \forall i \in \mathcal{V}_s \cup \mathcal{V}_t$ until it has seen 32 000 synthetic examples.

**Results.** We evaluate our trained model on the PASCALPF dataset (Ham et al., 2016) which consists of 1 351 image pairs within 20 classes, with the number of keypoints ranging from 4 to 17. Results of Hits@1 are shown in Table 3. Overall, our consensus architecture improves upon the state-of-the-art results of Zhang & Lee (2019) on almost all categories while our $L = 0$ baseline is weaker than the results reported in Zhang & Lee (2019), showing the benefits of applying our consensus stage. In addition, it shows that our method works also well even when not taking any visual information into account.

## 4.4 SEMI-SUPERVISED CROSS-LINGUAL KNOWLEDGE GRAPH ALIGNMENT

We evaluate our model on the DBP15K datasets (Sun et al., 2017) which link entities of the Chinese, Japanese and French knowledge graphs of DBPEDIA into the English version and vice versa. Each dataset contains exactly 15 000 links between equivalent entities, and we split those links into training and testing following upon previous works. For obtaining entity input features, we follow the experimental setup of Xu et al. (2019d): We retrieve monolingual FASTTEXT embeddings (Bojanowski et al., 2017) for each language separately, and align those into the same vector space afterwards (Lample et al., 2018). We use the sum of word embeddings as the final entity input representation (although more sophisticated approaches are just as conceivable).

**Architecture and parameters.** Our graph neural network operator mostly matches the one proposed in Xu et al. (2019d) where the direction of edges is retained, but not their specific relation type:

$$\vec{h}_i^{(t+1)} = \sigma \left( \boldsymbol{W}_1^{(t+1)} \vec{h}_i^{(t)} + \sum_{j \to i} \boldsymbol{W}_2^{(t+1)} \vec{h}_j^{(t)} + \sum_{i \to j} \boldsymbol{W}_3^{(t+1)} \vec{h}_j^{(t)} \right) \tag{9}$$

We use ReLU followed by dropout with probability 0.5 as our non-linearity $\sigma$, and obtain final node representations via $\vec{h}_i = \boldsymbol{W}_4[\vec{h}_i^{(1)}, \ldots, \vec{h}_i^{(T)}]$. We use a three-layer GNN ($T = 3$) both for obtaining initial similarities and for refining alignments with dimensionality 256 and 32, respectively. Training is performed using negative log likelihood in a semi-supervised fashion: For each training node $i$ in $\mathcal{V}_s$, we train $\mathcal{L}^{\text{(initial)}}$ sparsely by using the corresponding ground-truth node in $\mathcal{V}_t$, the top $k = 10$ entries in $\boldsymbol{S}_{i,:}$ and $k$ randomly sampled entities in $\mathcal{V}_t$. For the refinement phase, we update the sparse top $k$ correspondence matrix $L = 10$ times. For efficiency reasons, we train $\mathcal{L}^{\text{(initial)}}$ and $\mathcal{L}^{\text{(refined)}}$ sequentially for 100 epochs each.

**Results.** We report Hits@1 and Hits@10 to evaluate and compare our model to previous lines of work, see Table 4. In addition, we report results of a simple three-layer MLP which matches nodes purely based on initial word embeddings, and a variant of our model without the refinement of initial correspondences ($L = 0$). Our approach improves upon the state-of-the-art on all categories with

Table 4: Hits@1 (%) and Hits@10 (%) on the DBP15K dataset.

| Method | | ZH→EN @1 | ZH→EN @10 | EN→ZH @1 | EN→ZH @10 | JA→EN @1 | JA→EN @10 | EN→JA @1 | EN→JA @10 | FR→EN @1 | FR→EN @10 | EN→FR @1 | EN→FR @10 |
|---|---|---|---|---|---|---|---|---|---|---|---|---|---|
| GCN (Wang et al., 2018) | | 41.25 | 74.38 | 36.49 | 69.94 | 39.91 | 74.46 | 38.42 | 71.81 | 37.29 | 74.49 | 36.77 | 73.06 |
| BOOTEA (Sun et al., 2018) | | 62.94 | 84.75 | | | 62.23 | 85.39 | | | 65.30 | 87.44 | | |
| MUGNN (Cao et al., 2019) | | 49.40 | 84.40 | | | 50.10 | 85.70 | | | 49.60 | 87.00 | | |
| NAEA (Zhu et al., 2019) | | 65.01 | 86.73 | | | 64.14 | 87.27 | | | 67.32 | 89.43 | | |
| RDGCN (Wu et al., 2019) | | 70.75 | 84.55 | | | 76.74 | 89.54 | | | 88.64 | 95.72 | | |
| GMNN (Xu et al., 2019d) | | 67.93 | 78.48 | 65.28 | 79.64 | 73.97 | 87.15 | 71.29 | 84.63 | 89.38 | 95.25 | 88.18 | 94.75 |
| $\mathbf{\Psi}_{\theta_1} = $ MLP | $L = 0$ | 58.53 | 78.04 | 54.99 | 74.25 | 59.18 | 79.16 | 55.40 | 75.53 | 76.07 | 91.54 | 74.89 | 90.57 |
| **Ours** (sparse) | $L = 0$ | 67.59 | **87.47** | 64.38 | **83.56** | 71.95 | **89.74** | 68.88 | **86.84** | 83.36 | **96.03** | 82.16 | **95.28** |
| | $L = 10$ | **80.12** | **87.47** | **76.77** | **83.56** | **84.80** | **89.74** | **81.09** | **86.84** | **93.34** | **96.03** | **91.95** | **95.28** |

gains of up to 9.38 percentage points. In addition, our refinement strategy consistently improves upon the Hits@1 of initial correspondences by a significant margin, while results of Hits@10 are shared due to the refinement operating only on sparsified top 10 initial correspondences. Due to the scalability of our approach, we can easily apply a multitude of refinement iterations while still retaining large hidden feature dimensionalities.

## 5 LIMITATIONS

Our experimental results demonstrate that the proposed approach effectively solves challenging real-world problems. However, the expressive power of GNNs is closely related to the WL heuristic for graph isomorphism testing (Xu et al., 2019c; Morris et al., 2019), whose power and limitations are well understood (Arvind et al., 2015). Our method generally inherits these limitations. Hence, one possible limitation is that whenever two nodes are assigned the same color by WL, our approach may fail to converge to one of the possible solutions. For example, there may exist two nodes $i, j \in \mathcal{V}_t$ with equal neighborhood sets $\mathcal{N}_1(i) = \mathcal{N}_1(j)$. One can easily see that the feature matching procedure generates equal initial correspondence distributions $\boldsymbol{S}_{:,i}^{(0)} = \boldsymbol{S}_{:,j}^{(0)}$, resulting in the same mapped node indicator functions $\boldsymbol{I}_{|\mathcal{V}_s|}^\top \boldsymbol{S}_{:,i}^{(0)} = \boldsymbol{I}_{|\mathcal{V}_s|}^\top \boldsymbol{S}_{:,j}^{(0)}$ from $\mathcal{G}_s$ to nodes $i$ and $j$, respectively. Since both nodes share the same neighborhood, $\boldsymbol{\Psi}_{\theta_2}$ also produces the same distributed functions $\vec{o}_i^{(t)} = \vec{o}_j^{(t)}$. As a result, both column vectors $\hat{\boldsymbol{S}}_{:,i}^{(l)}$ and $\hat{\boldsymbol{S}}_{:,j}^{(l)}$ receive the same update, leading to non-convergence. In theory, one might resolve these ambiguities by adding a small amount of noise to $\hat{\boldsymbol{S}}^{(0)}$. However, the general amount of feature noise present in real-world datasets already ensures that this scenario is unlikely to occur.

## 6 RELATED WORK

Identifying correspondences between the nodes of two graphs has been studied in various domains and an extensive body of literature exists. Closely related problems are summarized under the terms *maximum common subgraph* (Kriege et al., 2019b), *network alignment* (Zhang, 2016), *graph edit distance* (Chen et al., 2019) and *graph matching* (Yan et al., 2016). We refer the reader to the Appendix F for a detailed discussion of the related work on these problems. Recently, graph neural networks have become a focus of research leading to various proposed *deep graph matching techniques* (Wang et al., 2019b; Zhang & Lee, 2019; Xu et al., 2019d; Derr et al., 2019). In Appendix G, we present a detailed overview of the related work in this field while highlighting individual differences and similarities to our proposed graph matching consensus procedure.

## 7 CONCLUSION

We presented a two-stage neural architecture for learning node correspondences between graphs in a supervised or semi-supervised fashion. Our approach is aimed towards reaching a neighborhood consensus between matchings, and can resolve violations of this criteria in an iterative fashion. In addition, we proposed enhancements to let our algorithm scale to large input domains. We evaluated our architecture on real-world datasets on which it consistently improved upon the state-of-the-art.

ACKNOWLEDGEMENTS

This work has been supported by the *German Research Association (DFG)* within the Collaborative Research Center SFB 876 *Providing Information by Resource-Constrained Analysis*, projects A6 and B2.

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

## A  Optimized Graph Matching Consensus Algorithm

Our final optimized algorithm is given in Algorithm 1:

---

**Algorithm 1** Optimized graph matching consensus algorithm

---

**Input**: $\mathcal{G}_s = (\mathcal{V}_s, \boldsymbol{A}_s, \boldsymbol{X}_s, \boldsymbol{E}_s)$, $\mathcal{G}_t = (\mathcal{V}_t, \boldsymbol{A}_t, \boldsymbol{X}_t, \boldsymbol{E}_t)$, hidden node dimensionality $d$, sparsity parameter $k$, number of consensus iterations $L$, number of random functions $r$

**Output**: Sparse soft correspondence matrix $\boldsymbol{S}^{(L)} \in [0,1]^{|\mathcal{V}_s| \times |\mathcal{V}_t|}$ with $k \cdot |\mathcal{V}_s|$ non-zero entries

---

$\boldsymbol{H}_s \leftarrow \boldsymbol{\Psi}_{\theta_1}(\boldsymbol{X}_s, \boldsymbol{A}_s, \boldsymbol{E}_s)$  $\triangleright$ Compute node embeddings $\boldsymbol{H}_s \in \mathbb{R}^{|\mathcal{V}_s| \times \cdot}$

$\boldsymbol{H}_t \leftarrow \boldsymbol{\Psi}_{\theta_1}(\boldsymbol{X}_s, \boldsymbol{A}_t, \boldsymbol{E}_t)$  $\triangleright$ Compute node embeddings $\boldsymbol{H}_t \in \mathbb{R}^{|\mathcal{V}_t| \times \cdot}$

$\hat{\boldsymbol{S}}^{(0)} \leftarrow \boldsymbol{H}_s \boldsymbol{H}_t^\top$  $\triangleright$ Local feature matching

$\hat{\boldsymbol{S}}^{(0)}_{i,:} \leftarrow \mathrm{top}_k(\hat{\boldsymbol{S}}^{(0)}_{i,:})$  $\triangleright$ Sparsify to top $k$ candidates $\forall i \in \{1, \dots, |\mathcal{V}_s|\}$

**for** $l$ in $\{1, \dots, L\}$ **do**  $\triangleright L \in \{L^{(\mathrm{train})}, L^{(\mathrm{test})}\}$

$\quad \boldsymbol{S}^{(l-1)}_{i,:} \leftarrow \mathrm{softmax}(\hat{\boldsymbol{S}}^{(l-1)}_{i,:})$  $\triangleright$ Normalize scores $\forall i \in \{1, \dots, |\mathcal{V}_s|\}$

$\quad \boldsymbol{R}^{(l)}_s \sim \mathcal{N}(0,1)$  $\triangleright$ Sample random node function $\boldsymbol{R}^{(l)}_s \in \mathbb{R}^{|\mathcal{V}_s| \times r}$

$\quad \boldsymbol{R}^{(l)}_t \leftarrow \boldsymbol{S}^\top_{(l-1)} \boldsymbol{R}^{(l)}_s$  $\triangleright$ Map random node functions $\boldsymbol{R}^{(l)}_s$ from $\mathcal{G}_s$ to $\mathcal{G}_t$

$\quad \boldsymbol{O}_s \leftarrow \boldsymbol{\Psi}_{\theta_2}(\boldsymbol{R}^{(l)}_s, \boldsymbol{A}_s, \boldsymbol{E}_s)$  $\triangleright$ Distribute function $\boldsymbol{R}^{(l)}_s$ on $\mathcal{G}_s$

$\quad \boldsymbol{O}_t \leftarrow \boldsymbol{\Psi}_{\theta_2}(\boldsymbol{R}^{(l)}_t, \boldsymbol{A}_t, \boldsymbol{E}_t)$  $\triangleright$ Distribute function $\boldsymbol{R}^{(l)}_t$ on $\mathcal{G}_t$

$\quad \vec{d}_{i,j} \leftarrow \vec{o}^{(s)}_i - \vec{o}^{(t)}_j$  $\triangleright$ Compute neighborhood consensus measure

$\quad \hat{S}^{(l)}_{i,j} \leftarrow \hat{S}^{(l-1)}_{i,j} + \Phi_{\theta_3}(\vec{d}_{i,j})$  $\triangleright$ Perform trainable correspondence update

**end for**

$\boldsymbol{S}^{(L)}_{i,:} \leftarrow \mathrm{softmax}(\hat{\boldsymbol{S}}^{(L)}_{i,:})$  $\triangleright$ Normalize scores $\forall i \in \{1, \dots, |\mathcal{V}_s|\}$

**return** $\boldsymbol{S}^{(L)}$

---

## B  Proof for Theorem 1

*Proof.* Since $\boldsymbol{\Psi}_{\theta_2}$ is permutation equivariant, it holds for any node feature matrix $\boldsymbol{X}_s \in \mathbb{R}^{|\mathcal{V}_s| \times \cdot}$ that $\boldsymbol{\Psi}_{\theta_2}(\boldsymbol{S}^\top \boldsymbol{X}_s, \boldsymbol{S}^\top \boldsymbol{A}_s \boldsymbol{S}) = \boldsymbol{S}^\top \boldsymbol{\Psi}_{\theta_2}(\boldsymbol{X}_s, \boldsymbol{A}_s)$. With $\boldsymbol{X}_t = \boldsymbol{S}^\top \boldsymbol{X}_s$ and $\boldsymbol{A}_t = \boldsymbol{S}^\top \boldsymbol{A}_s \boldsymbol{S}$, it follows that

$$\boldsymbol{O}_t = \boldsymbol{\Psi}_{\theta_2}(\boldsymbol{X}_t, \boldsymbol{A}_t) = \boldsymbol{\Psi}_{\theta_2}(\boldsymbol{S}^\top \boldsymbol{X}_s, \boldsymbol{S}^\top \boldsymbol{A}_s \boldsymbol{S}) = \boldsymbol{S}^\top \boldsymbol{\Psi}_{\theta_2}(\boldsymbol{X}_s, \boldsymbol{A}_s) = \boldsymbol{S}^\top \boldsymbol{O}_s.$$

Hence, it shows that $\vec{o}^{(s)}_i = (\boldsymbol{S}^\top \boldsymbol{O}_s)_{\pi(i)} = \vec{o}^{(t)}_{\pi(i)}$ for any node $i \in \mathcal{V}_s$, resulting in $\vec{d}_{i,\pi(i)} = \vec{0}$. $\quad \square$

## C  Proof for Theorem 2

*Proof.* Let be $\vec{d}_{i,j} = \vec{o}^{(s)}_i - \vec{o}^{(t)}_j = \vec{0}$. Then, the $T$-layered GNN $\boldsymbol{\Psi}_{\theta_2}$ maps both $T$-hop neighborhoods around nodes $i \in \mathcal{V}_s$ and $j \in \mathcal{V}_t$ to the same vectorial representation:

$$\vec{o}^{(s)}_i = \boldsymbol{\Psi}_{\theta_2}(\boldsymbol{I}^{|\mathcal{V}_s|}_{\mathcal{N}_T(i),:}, \boldsymbol{A}^{(s)}_{\mathcal{N}_T(i), \mathcal{N}_T(i)})_i = \boldsymbol{\Psi}_{\theta_2}((\boldsymbol{S}^\top \boldsymbol{I}_{|\mathcal{V}_s|})_{\mathcal{N}_T(j),:}, \boldsymbol{A}^{(t)}_{\mathcal{N}_T(j), \mathcal{N}_T(j)})_j = \vec{o}^{(t)}_j \quad (10)$$

Because $\boldsymbol{\Psi}_{\theta_2}$ is as powerful as the WL heuristic in distinguishing graph structures (Xu et al., 2019c; Morris et al., 2019) and is operating on injective node colorings $\boldsymbol{I}_{|\mathcal{V}|_s}$, it has the power to distinguish *any* graph structure from $\mathcal{G}_s[\mathcal{N}_T(i)] = (\mathcal{N}_T(i), \boldsymbol{I}^{|\mathcal{V}_s|}_{\mathcal{N}_T(i),:}, \boldsymbol{A}^{(s)}_{\mathcal{N}_T(i), \mathcal{N}_T(i)})$, *cf.* (Murphy et al., 2019). Since $\vec{o}^{(s)}_i$ holds information about every node in $\mathcal{G}_s[\mathcal{N}_T(i)]$, it necessarily holds that $\mathcal{G}_s[\mathcal{N}_T(i)] \simeq \mathcal{G}_t[\mathcal{N}_T(j)]$ in case $\vec{o}^{(s)}_i = \vec{o}^{(t)}_j$, where $\simeq$ denotes the labeled graph isomorphism relation. Hence, there exists an isomorphism $\boldsymbol{P} \in \{0,1\}^{|\mathcal{N}_T(i)| \times |\mathcal{N}_T(j)|}$ between $\mathcal{G}_s[\mathcal{N}_T(j)]$ and $\mathcal{G}_t[\mathcal{N}_T(j)]$ such that

$$\boldsymbol{I}^{|\mathcal{V}_s|}_{\mathcal{N}_T(i),:} = \boldsymbol{P}(\boldsymbol{S}^\top \boldsymbol{I}_{|\mathcal{V}_s|})_{\mathcal{N}_T(j),:} \quad \text{and} \quad \boldsymbol{A}^{(s)}_{\mathcal{N}_T(i), \mathcal{N}_T(i)} = \boldsymbol{P} \boldsymbol{A}^{(t)}_{\mathcal{N}_T(j), \mathcal{N}_T(j)} \boldsymbol{P}^\top \quad (11)$$

With $\boldsymbol{I}_{|\mathcal{V}_s|}$ being the identity matrix, it follows that $\boldsymbol{I}^{|\mathcal{V}_s|}_{\mathcal{N}_T(i),:} = \boldsymbol{P} \boldsymbol{S}^\top_{\mathcal{N}_T(j),:}$. Furthermore, it holds that $\boldsymbol{I}^{|\mathcal{V}_s|}_{\mathcal{N}_T(i), \mathcal{N}_T(i)} = \boldsymbol{P} \boldsymbol{S}^\top_{\mathcal{N}_T(j), \mathcal{N}_T(i)}$ when reducing $\boldsymbol{I}^{|\mathcal{V}_s|}_{\mathcal{N}_T(i),:}$ to its column-wise non-zero entries. It follows that $\boldsymbol{S}_{\mathcal{N}_T(i), \mathcal{N}_T(j)} = \boldsymbol{P}$ is a permutation matrix describing an isomorphism.

Table 5: Hits@1 (%) on the PASCALVOC dataset with Berkeley keypoint annotations.

| Method | | Aero | Bike | Bird | Boat | Bottle | Bus | Car | Cat | Chair | Cow | Table | Dog | Horse | M-Bike | Person | Plant | Sheep | Sofa | Train | TV | Mean |
|---|---|---|---|---|---|---|---|---|---|---|---|---|---|---|---|---|---|---|---|---|---|---|
| isotropic | $L=0$ | 44.3 | 62.0 | 48.4 | 53.9 | 73.3 | 80.4 | 72.2 | 64.2 | 30.3 | 52.7 | 79.4 | 56.6 | 62.3 | 56.2 | 47.5 | 74.0 | 59.8 | 79.9 | 81.9 | 83.0 | 63.1 |
| $\Psi_{\theta_2}=AX$ | $L=10$ | 45.9 | 60.5 | 49.0 | 59.7 | 72.8 | 80.9 | 77.4 | 67.2 | 34.1 | 56.3 | 80.4 | 59.5 | 68.6 | 53.9 | 48.6 | 75.5 | 60.8 | 91.5 | 84.8 | 80.3 | 65.4 |
| | $L=20$ | 44.7 | 61.5 | 53.0 | 63.1 | 73.6 | 81.2 | 75.2 | 68.1 | 33.9 | 57.1 | 80.5 | 59.7 | 66.5 | 54.4 | 51.6 | 74.9 | 63.6 | 85.4 | 79.6 | 82.3 | 65.5 |
| $\Psi_{\theta_2}=$ GNN | $L=10$ | 46.5 | 63.7 | 54.9 | 60.9 | 79.4 | **84.1** | 76.4 | 68.3 | **38.5** | **61.5** | **80.6** | 59.7 | 69.8 | 58.4 | 54.3 | 76.4 | 64.5 | **95.7** | **87.9** | 81.3 | 68.1 |
| | $L=20$ | **50.1** | **65.4** | **55.7** | **65.3** | **80.0** | 83.5 | **78.3** | **69.7** | 34.7 | 60.7 | 70.4 | **59.9** | **70.0** | **62.2** | **56.1** | **80.2** | **70.3** | 88.8 | 81.1 | **84.3** | **68.3** |

Table 6: Hits@1 (%) on the DBP15K dataset.

| **Method** | | **ZH→EN** | **EN→ZH** | **JA→EN** | **EN→JA** | **FR→EN** | **EN→FR** |
|---|---|---|---|---|---|---|---|
| | $L=0$ | 67.59 | 64.38 | 71.95 | 68.88 | 83.36 | 82.16 |
| $\Psi_{\theta_2}=AX$ | $L=10$ | 71.61 | 68.52 | 77.18 | 76.53 | 85.69 | 85.96 |
| $\Psi_{\theta_2}=$ GNN | $L=10$ | **80.12** | **76.77** | **84.80** | **81.09** | **93.34** | **91.95** |

Moreover, if $\vec{d}_{i,\operatorname{argmax} \boldsymbol{S}_{i,:}} = \vec{0}$ for all $i \in \mathcal{V}_s$, it directly follows that $\boldsymbol{S}$ is holding submatrices describing isomorphisms between any $T$-hop subgraphs around $i \in \mathcal{V}_s$ and $\operatorname{argmax} \boldsymbol{S}_{i,:} \in \mathcal{V}_t$. Assume there exists nodes $i, i' \in \mathcal{V}_s$ that map to the same node $j = \operatorname{argmax} \boldsymbol{S}_{i,:} = \operatorname{argmax} \boldsymbol{S}_{i',:} \in \mathcal{V}_t$. It follows that $\vec{o}_i^{(s)} = \vec{o}_j^{(t)} = \vec{o}_{i'}^{(s)}$ which contradicts the injectivity requirements of $\text{AGGREGATE}^{(t)}$ and $\text{UPDATE}^{(t)}$ for all $t \in \{1, \ldots, T\}$. Hence, $\boldsymbol{S}$ must be itself a permutation matrix describing an isomorphism between $\mathcal{G}_s$ and $\mathcal{G}_t$. □

## D    COMPARISON TO THE GRADUATED ASSIGNMENT ALGORITHM

As stated in Section 3.3, our algorithm can be viewed as a generalization of the graduated assignment algorithm (Gold & Rangarajan, 1996) extending it by trainable parameters. To evaluate the impact of a trainable refinement procedure, we replicated the experiments of Sections 4.2 and 4.4 by implementing $\Psi_{\theta_2}$ via a non-trainable, one-layer GNN instantiation $\Psi_{\theta_2}(\boldsymbol{X}, \boldsymbol{A}, \boldsymbol{E}) = \boldsymbol{AX}$.

The results in Tables 5 and 6 show that using trainable neural networks $\Psi_{\theta_2}$ consistently improves upon the results of using the fixed-function message passing scheme. While it is difficult to encode meaningful similarities between node and edge features in a fixed-function pipeline, our approach is able to *learn* how to make use of those features to guide the refinement procedure further. In addition, it allows us to choose from a variety of task-dependent GNN operators, *e.g.*, for learning geometric/edge conditioned patterns or for fulfilling injectivity requirements. The theoretical expressivity discussed in Section 5 could even be enhanced by making use of higher-order GNNs, which we leave for future work.

## E    ROBUSTNESS TOWARDS NODE ADDITION OR REMOVAL

To experimentally validate the robustness of our approach towards node addition (or removal), we conducted additional synthetic experiments in a similar fashion to Xu et al. (2019b). We form graph-pairs by treating an Erdős & Rényi graph with $|\mathcal{V}_s| \in \{50, 100\}$ nodes and edge probability $p \in \{0.1, 0.2\}$ as our source graph $\mathcal{G}_s$. The target graph $\mathcal{G}_t$ is then constructed by first adding $q\%$ noisy nodes to the source graph, *i.e.*, $|\mathcal{V}_t| = (1 + q\%)|\mathcal{V}_s|$, and generating edges between these nodes and all other nodes based on the edge probability $p$ afterwards. We use the same network architecture and training procedure as described in Section 4.1.

Figure 3 visualizes the Hits@1 for different choices of $|\mathcal{V}_s|$, $p$ and $q \in \{0.0, 0.1, 0.2, 0.3, 0.4, 0.5\}$. As one can see, our consensus stage is extremely robust to the addition or removal of nodes while the first stage alone has major difficulties in finding the right matching. This can be explained by the fact that unmatched nodes do not have any influence on the neighborhood consensus error since those nodes do not obtain a color from the functional map given by $\boldsymbol{S}$. Our neural architecture is able to detect and gradually decrease any false positive influence of these nodes in the refinement stage.

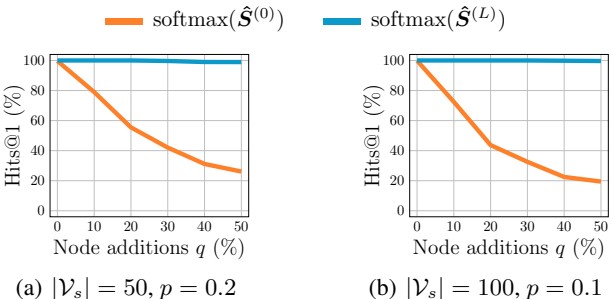

Figure 3: The performance of our method on synthetic data with node additions.

## F  RELATED WORK I

Identifying correspondences between the nodes of two graphs is a problem arising in various domains and has been studied under different terms. In graph theory, the combinatorial *maximum common subgraph isomorphism* problem is studied, which asks for the largest graph that is contained as subgraph in two given graphs. The problem is NP-hard in general and remains so even in trees (Garey & Johnson, 1979) unless the common subgraph is required to be connected (Matula, 1978). Moreover, most variants of the problem are difficult to approximate with theoretical guarantees (Kann, 1992). We refer the reader to the survey by Kriege et al. (2019b) for a overview of the complexity results noting that exact polynomial-time algorithms are available for specific problem variants only that are most relevant in cheminformatics.

Fundamentally different techniques have been developed in bioinformatics and computer vision, where the problem is commonly referred to as *network alignment* or *graph matching*. In these areas large networks without any specific structural properties are common and the studied techniques are non-exact. In graph matching, for two graphs of order $n$ with adjacency matrix $\boldsymbol{A}_s$ and $\boldsymbol{A}_t$, respectively, typically the function

$$\left\| \boldsymbol{A}_s - \boldsymbol{S}^\top \boldsymbol{A}_t \boldsymbol{S} \right\|_F^2 = \left\| \boldsymbol{A}_s \right\|_F^2 + \left\| \boldsymbol{A}_t \right\|_F^2 - 2 \sum_{\substack{i,i' \in \mathcal{V}_s \\ j,j' \in \mathcal{V}_t}} A_{i,i'}^{(s)} A_{j,j'}^{(t)} S_{i,j} S_{i',j'} \tag{12}$$

is to be minimized, where $\boldsymbol{S} \in \mathcal{P}$ with $\mathcal{P}$ the set of $n \times n$ permutation matrices and $\left\| \boldsymbol{A} \right\|_F^2 = \sum_{i,i' \in \mathcal{V}} A_{i,i'}^2$ denotes the squared Frobenius norm. Since the first two terms of the right-hand side do not depend on $\boldsymbol{S}$, minimizing Equation (12) is equivalent in terms of optimal solutions to the problem of Equation (1). We briefly summarize important related work in graph matching and refer the reader to the recent survey by Yan et al. (2016) for a more detailed discussion. There is a long line of research trying to minimize Equation (12) for $\boldsymbol{S} \in [0,1]^{n \times n}$ by a Frank-Wolfe type algorithm (Jaggi, 2013) and finally projecting the fractional solution to $\mathcal{P}$ (Gold & Rangarajan, 1996; Zaslavskiy et al., 2009; Leordeanu et al., 2009; Egozi et al., 2013; Zhou & De la Torre, 2016). However, the applicability of relaxation and projection is still poorly understood and only few theoretical results exist (Aflalo et al., 2015; Lyzinski et al., 2016). A classical result by Tinhofer (1991) states that the WL heuristic distinguishes two graphs $\mathcal{G}_s$ and $\mathcal{G}_t$ if and only if there is no fractional $\boldsymbol{S}$ such that the objective function in Equation (12) takes 0. Kersting et al. (2014) showed how the Frank-Wolfe algorithm can be modified to obtain the WL partition. Aflalo et al. (2015) proved that the standard relaxation yields a correct solution for a particular class of asymmetric graphs, which can be characterized by the spectral properties of their adjacency matrix. Finally, Bento & Ioannidis (2018) studied various relaxations, their complexity and properties. Other approaches to graph matching exist, *e.g.*, based on spectral relaxations (Umeyama, 1988; Leordeanu & Hebert, 2005) or random walks (Gori et al., 2005). The problem of graph matching is closely related to the notoriously hard quadratic assignment problem (QAP) (Zhou & De la Torre, 2016), which has been studied in operations research for decades. Equation (1) can be directly interpreted as *Koopmans-Beckmann's QAP*. The more recent literature on graph matching typically considers a weighted version, where node and edge similarities are taken into account. This leads to the formulation as *Lawler's QAP*, which involves an affinity matrix of size $n^2 \times n^2$ and is computational demanding.

Zhou & De la Torre (2016) proposed to factorize the affinity matrix into smaller matrices and incorporated global geometric constraints. Zhang et al. (2019c) studied *kernelized graph matching*, where the node and edge similarities are kernels, which allows to express the graph matching problem again as Koopmans-Beckmann's QAP in the associated Hilbert space. Inspired by established methods for Maximum-A-Posteriori (MAP) inference in conditional random fields, Swoboda et al. (2017) studied several Lagrangean decompositions of the graph matching problem, which are solved by dual ascent algorithms also known as *message passing*. Specific message passing schedules and update mechanisms leading to state-of-the-art performance in graph matching tasks have been identified experimentally. Recently, *functional representation* for graph matching has been proposed as a generalizing concept with the additional goal to avoid the construction of the affinity matrix (Wang et al., 2019a).

**Graph edit distance.**  A related concept studied in computer vision is the *graph edit distance*, which measures the minimum cost required to transform a graph into another graph by adding, deleting and substituting vertices and edges. The idea has been proposed for pattern recognition tasks more than 30 years ago (Sanfeliu & Fu, 1983). However, its computation is NP-hard, since it generalizes the maximum common subgraph problem (Bunke, 1997). Moreover, it is also closely related to the quadratic assignment problem (Bougleux et al., 2017). Recently several elaborated exact algorithms for computing the graph edit distance have been proposed (Gouda & Hassaan, 2016; Lerouge et al., 2017; Chen et al., 2019), but are still limited to small graphs. Therefore, heuristics based on the assignment problem have been proposed (Riesen & Bunke, 2009) and are widely used in practice (Stauffer et al., 2017). The original approach requires cubic running time, which can be reduced to quadratic time using greedy strategies (Riesen et al., 2015a;b), and even linear time for restricted cost functions (Kriege et al., 2019a).

**Network alignment.**  The problem of *network alignment* typically is defined analogously to Equation (1), where in addition a similarity function between pairs of nodes is given. Most algorithms follow a two step approach: First, an $n \times n$ node-to-node similarity matrix $M$ is computed from the given similarity function and the topology of the two graphs. Then, in the second step, an alignment is computed by solving the assignment problem for $M$. Singh et al. (2008) proposed IsoRank, which is based on the adjacency matrix of the product graph $K = A_s \otimes A_t$ of $\mathcal{G}_s$ and $\mathcal{G}_t$, where $\otimes$ denotes the Kronecker product. The matrix $M$ is obtained by applying PageRank (Page et al., 1999) using a normalized version of $K$ as the Google matrix and the node similarities as the personalization vector. Kollias et al. (2012) proposed an efficient approximation of IsoRank by decomposition techniques to avoid generating the product graph of quadratic size. Zhang (2016) present an extension supporting vertex and edge similarities and propose its computation using non-exact techniques. Klau (2009) proposed to solve network alignment by linearizing the quadratic optimization problem to obtain an integer linear program, which is then approached via Lagrangian relaxation. Bayati et al. (2013) developed a message passing algorithm for sparse network alignment, where only a small number of matches between the vertices of the two graphs are allowed.

The techniques briefly summarized above aim to find an optimal correspondence according to a clearly defined objective function. In practical applications, it is often difficult to specify node and edge similarity functions. Recently, it has been proposed to *learn* such functions for a specific task, *e.g.*, in form of a cost model for the graph edit distance (Cortés et al., 2019). A more principled approach has been proposed by Caetano et al. (2009) where the goal is to learn correspondences.

# G  RELATED WORK II

The method presented in this work is related to different lines of research. Deep graph matching procedures have been investigated from multiple perspectives, *e.g.*, by utilizing local node feature matchings and cross-graph embeddings (Li et al., 2019). The idea of refining local feature matchings by enforcing neighborhood consistency has been relevant for several years for matching in images (Sattler et al., 2009). Furthermore, the functional maps framework aims to solve a similar problem for manifolds (Halimi et al., 2019).

**Deep graph matching.**  Recently, the problem of graph matching has been heavily investigated in a deep fashion. For example, Zanfir & Sminchisescu (2018); Wang et al. (2019b); Zhang & Lee

(2019) develop supervised deep graph matching networks based on displacement and combinatorial objectives, respectively. Zanfir & Sminchisescu (2018) model the graph matching affinity via a differentiable, but unlearnable spectral graph matching solver (Leordeanu & Hebert, 2005). In contrast, our matching procedure is fully-learnable. Wang et al. (2019b) use node-wise features in combination with dense node-to-node cross-graph affinities, distribute them in a local fashion, and adopt SINKHORN normalization for the final task of linear assignment. Zhang & Lee (2019) propose a compositional message passing algorithm that maps point coordinates into a high-dimensional space. The final matching procedure is done by computing the pairwise inner product between point embeddings. However, neither of these approaches can naturally resolve violations of inconsistent neighborhood assignments as we do in our work.

Xu et al. (2019b) tackles the problem of graph matching by relating it to the Gromov-Wasserstein discrepancy (Peyré et al., 2016). In addition, the optimal transport objective is enhanced by simultaneously learning node embeddings which shall account for the noise in both graphs. In a follow-up work, Xu et al. (2019a) extend this concept to the tasks of multi-graph partioning and matching by learning a Gromov-Wasserstein barycenter. Our approach also resembles the optimal transport between nodes, but works in a supervised fashion for sets of graphs and is therefore able to generalize to unseen graph instances.

In addition, the task of network alignment has been recently investigated from multiple perspectives. Derr et al. (2019) leverage CYCLEGANs (Zhu et al., 2017) to align NODE2VEC embeddings (Grover & Leskovec, 2016) and find matchings based on the nearest neighbor in the embedding space. Zhang et al. (2019a) design a deep graph model based on global and local network topology preservation as auxiliary tasks. Heimann et al. (2018) utilize a fast, but purely local and greedy matching procedure based on local node embedding similarity.

Furthermore, Bai et al. (2019) use shared graph neural networks to approximate the graph edit distance between two graphs. Here, a (non-differentiable) histogram of correspondence scores is used to fine-tune the output of the network. In a follow-up work, Bai et al. (2018) proposed to order the correspondence matrix in a breadth-first-search fashion and to process it further with the help of traditional CNNs. Both approaches only operate on local node embeddings, and are hence prone to match correspondences inconsistently.

**Intra- and inter-graph message passing.** The concept of enhancing intra-graph node embeddings by inter-graph node embeddings has been already heavily investigated in practice (Li et al., 2019; Wang et al., 2019b; Xu et al., 2019d). Li et al. (2019) and Wang et al. (2019b) enhance the GNN operator by not only aggregating information from local neighbors, but also from similar embeddings in the other graph by utilizing a cross-graph matching procedure. Xu et al. (2019d) leverage alternating GNNs to propagate local features of one graph throughout the second graph. Wang & Solomon (2019) tackle the problem of finding an unknown rigid motion between point clouds by relating it to a point cloud matching problem followed by a differentiable SVD module. Intra-graph node embeddings are passed via a Transformer module before feature matching based on inner product similarity scores takes place. However, neither of these approaches is designed to achieve a consistent matching, due to only operating on localized node embeddings which are alone not sufficient to resolve ambiguities in the matchings. Nonetheless, we argue that these methods can be used to strengthen the initial feature matching procedure, making our approach orthogonal to improvements in this field.

**Neighborhood consensus for image matching.** Methods to obtain consistency of correspondences in local neighborhoods have a rich history in computer vision, dating back several years (Sattler et al., 2009; Sivic & Zisserman, 2003; Schmid & Mohr, 1997). They are known for heavily improving results of local feature matching procedures while being computational efficient. Recently, a deep neural network for neighborhood consensus using 4D convolution was proposed (Rocco et al., 2018). While it is related to our method, the 4D convolution can not be efficiently transferred to the graph domain directly, since it would lead to applying a GNN on the product graph with $\mathcal{O}(n^2)$ nodes and $\mathcal{O}(n^4)$ edges. Our algorithm also infers errors for the (sparse) product graph but performs the necessary computations on the original graphs.

**Functional maps.** The functional maps framework was proposed to provide a way to define continuous maps between function spaces on manifolds and is commonly applied to solve the task of

Table 7: Statistics of the WILLOW-OBJECTCLASS dataset.

| Category | Graphs | Keypoints | Edges |
|---|---|---|---|
| **Face** | 108 | 10 | $21 - 22$ |
| **Motorbike** | 40 | 10 | $21 - 22$ |
| **Car** | 40 | 10 | $18 - 21$ |
| **Duck** | 50 | 10 | $19 - 21$ |
| **Winebottle** | 66 | 10 | $19 - 22$ |

Table 8: Statistics of the DBP15K dataset.

| Datasets | | Entities | Relation types | Relations |
|---|---|---|---|---|
| **ZH↔EN** | Chinese | 19 388 | 1 701 | 70 414 |
| | English | 19 572 | 3 024 | 95 142 |
| **JA↔EN** | Japanese | 19 814 | 1 299 | 77 214 |
| | English | 19 780 | 2 452 | 93 484 |
| **FR↔EN** | French | 19 661 | 903 | 105 998 |
| | English | 19 993 | 2 111 | 115 722 |

Table 9: Statistics of the PASCALVOC dataset with Berkeley annotations.

| Category | Train graphs | Test graphs | Keypoints | Edges | Category | Train graphs | Test graphs | Keypoints | Edges |
|---|---|---|---|---|---|---|---|---|---|
| **Aeroplane** | 468 | 136 | $1 - 16$ | $0 - 41$ | **Diningtable** | 27 | 5 | $2 - 8$ | $2 - 8$ |
| **Bicycle** | 210 | 53 | $2 - 11$ | $1 - 26$ | **Dog** | 608 | 147 | $1 - 16$ | $0 - 41$ |
| **Bird** | 613 | 117 | $1 - 12$ | $0 - 30$ | **Horse** | 217 | 45 | $2 - 16$ | $1 - 38$ |
| **Boat** | 411 | 88 | $1 - 11$ | $0 - 25$ | **Motorbike** | 234 | 60 | $1 - 10$ | $0 - 23$ |
| **Bottle** | 466 | 120 | $1 - 8$ | $0 - 17$ | **Person** | 539 | 156 | $4 - 19$ | $5 - 49$ |
| **Bus** | 288 | 52 | $1 - 8$ | $0 - 17$ | **Pottedplant** | 429 | 99 | $1 - 6$ | $0 - 11$ |
| **Car** | 522 | 160 | $1 - 13$ | $0 - 27$ | **Sheep** | 338 | 73 | $1 - 16$ | $0 - 39$ |
| **Cat** | 415 | 101 | $3 - 16$ | $3 - 40$ | **Sofa** | 73 | 8 | $2 - 12$ | $1 - 27$ |
| **Chair** | 298 | 63 | $1 - 10$ | $0 - 23$ | **Train** | 166 | 43 | $1 - 6$ | $0 - 10$ |
| **Cow** | 257 | 55 | $1 - 16$ | $0 - 40$ | **TV Monitor** | 374 | 90 | $1 - 8$ | $0 - 17$ |

3D shape correspondence (Ovsjanikov et al., 2012; Litany et al., 2017; Rodolà et al., 2017; Halimi et al., 2019). Recently, a similar approach was presented to find functional correspondences between graph function spaces (Wang et al., 2019a). The functional map is established by using a low-dimensional basis representation, *e.g.*, the eigenbasis of the graph Laplacian as generalized Fourier transform. Since the basis is usually truncated to the $k$ vectors with the largest eigenvalues, these approaches focus on establishing global correspondences. However, such global methods have the inherent disadvantage that they often fail to find partial matchings due to the domain-dependent eigenbasis. Furthermore, the basis computation has to be approximated in order to scale to large inputs.

## H    DATASET STATISTICS

We give detailed descriptions of all datasets used in our experiments, *cf.* Tables 7, 8, 9 and 10.

## I    QUALITATIVE KEYPOINT MATCHING RESULTS

Figure 4 visualizes qualitative examples from the task of keypoint matching on the WILLOW-OBJECTCLASS dataset. Examples were selected as follows: Figure 4(a), (b) and (c) show examples where the initial feature matching procedure fails, but where our refinement procedure is able to recover *all* correspondences succesfully. Figure 4(d) visualizes a rare failure case. However, while the initial feature matching procedure maps most of the keypoints to the same target keypoint, our refinement strategy is still able to succesfully resolve this violation. In addition, note that the target image contains wrong labels, *e.g.*, the eye of the duck, so that some keypoint mappings are mistakenly considered to be wrong.

Table 10: Statistics of the PASCALPF dataset.

| Category | Graphs | Pairs | Keypoints | Edges | Category | Graphs | Pairs | Keypoints | Edges |
|---|---|---|---|---|---|---|---|---|---|
| **Aeroplane** | 69 | 69 | $9 - 16$ | $72 - 128$ | **Diningtable** | 38 | 38 | $4 - 8$ | $12 - 56$ |
| **Bicycle** | 133 | 133 | $10 - 11$ | $80 - 88$ | **Dog** | 106 | 106 | $5 - 15$ | $20 - 120$ |
| **Bird** | 50 | 50 | $5 - 9$ | $20 - 72$ | **Horse** | 39 | 39 | $5 - 16$ | $20 - 128$ |
| **Boat** | 28 | 28 | $4 - 9$ | $12 - 72$ | **Motorbike** | 120 | 120 | $5 - 10$ | $20 - 80$ |
| **Bottle** | 42 | 42 | $4 - 8$ | $12 - 56$ | **Person** | 56 | 56 | $10 - 17$ | $80 - 136$ |
| **Bus** | 140 | 140 | $4 - 8$ | $12 - 56$ | **Pottedplant** | 35 | 35 | $4 - 6$ | $12 - 30$ |
| **Car** | 84 | 84 | $4 - 11$ | $12 - 88$ | **Sheep** | 6 | 6 | $5 - 7$ | $20 - 42$ |
| **Cat** | 119 | 119 | $5 - 16$ | $20 - 128$ | **Sofa** | 59 | 59 | $4 - 10$ | $12 - 80$ |
| **Chair** | 59 | 59 | $4 - 10$ | $12 - 80$ | **Train** | 88 | 88 | $4 - 5$ | $12 - 20$ |
| **Cow** | 15 | 15 | $5 - 16$ | $20 - 128$ | **TV Monitor** | 65 | 65 | $4 - 6$ | $12 - 30$ |

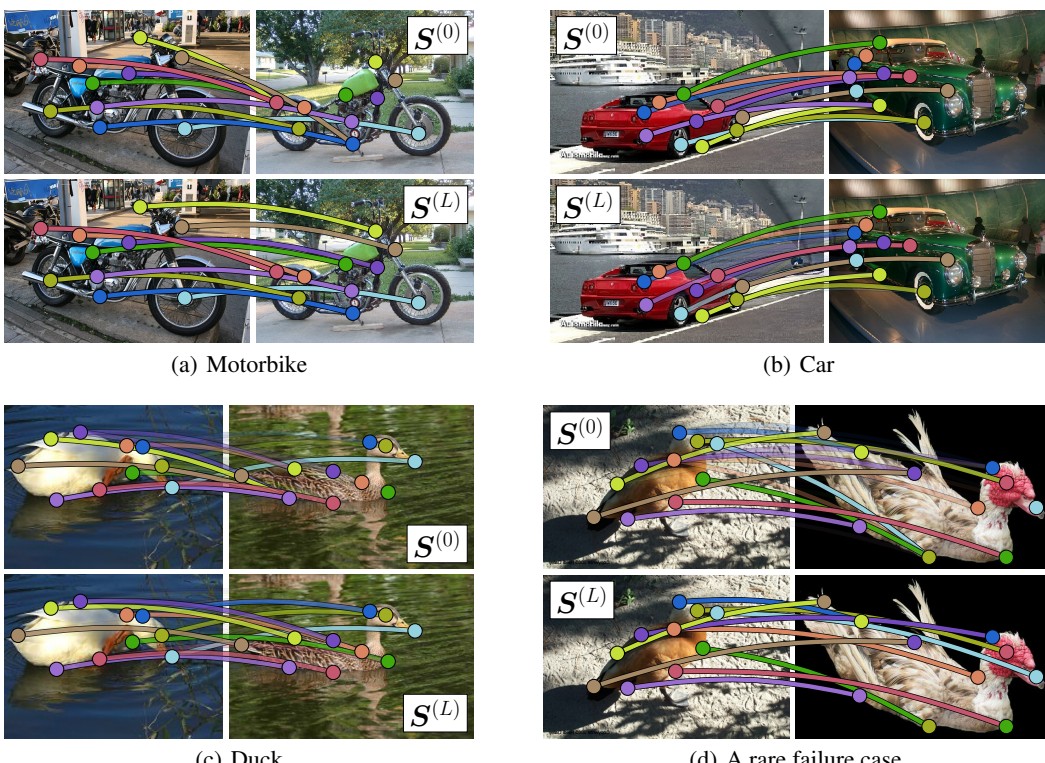

(a) Motorbike

(b) Car

(c) Duck

(d) A rare failure case

Figure 4: Qualitative examples from the WILLOW-OBJECTCLASS dataset. Images on the left represent the source, whereas images on the right represent the target. For each example, we visualize both the result of the initial feature matching procedure $S^{(0)}$ (top) and the result obtained after refinement $S^{(L)}$ (bottom).

