# OpenReview forum: "Deep Graph Matching Consensus"
_ICLR.cc/2020/Conference — Accept (Poster)_

### Official Review · AnonReviewer3 · 2019-10-23
**Official Blind Review #3**

**Rating:** 6

**Review:**

This paper suggests a framework for answering graph matching questions consisting of local node embeddings with a message passing refinement step.

The paper has well written text, offers what appear to be nice experiments validating the method and discusses its own limitations.

I am giving a weak accept.

I the weak accept is my reflection of my inability to provide useful feedback.  This is also out of domain for me and I am not a useful reviewer for this paper.

As a general comment, I will say that the paper adopts a highly mathematical style that will be off putting to many readers.  Many of the expressions are 'high entropy'  For instance, the embedding network is given as $\mathbf{\Psi}_{\theta_1}$ throughout.  This is seven levels of typographical distinction for the main character of the story.  This is a (1) bold (2) capital (3) Greek letter with a (4) subscript that is (5) greek with a (6) subscript that is (7) numeric.  I understand that each of these levels of distinction has a purpose and a meaning, but it is also arguably a much richer designation that is necessary.  Generally the paper feels like this, as if its being too unnecessarily specific.  Personally I found the paper rather difficult to read and decompose, which I believe does count against the paper.  The overly specific nature of paper will cut into its potential readership strongly.

**Experience Assessment:**

I do not know much about this area.

**Review Assessment: Checking Correctness Of Derivations And Theory:**

I did not assess the derivations or theory.

**Review Assessment: Checking Correctness Of Experiments:**

I did not assess the experiments.

**Review Assessment: Thoroughness In Paper Reading:**

I read the paper at least twice and used my best judgement in assessing the paper.

---

> ### Author Response · Authors · 2019-11-11
> **Response to Review #3**
>
> Dear Reviewer,
>
> thanks a lot for reviewing our paper and for the positive feedback.
>
> To precisely describe our architecture with the theorems and their proofs, we opt for introducing a rigorous mathematical notation. In addition, we wanted to describe our procedure as general as possible. For example, $\Psi_{\theta}$ is not limited to be a particular GNN instance but can be any trainable architecture outputting node embeddings. We will strengthen the intuitive explanations and simplify notation in a revised version.

---

### Official Review · AnonReviewer1 · 2019-10-26
**Official Blind Review #1**

**Rating:** 6

**Review:**

This paper proposes a two-stage GNN-based architecture to establish correspondences between two graphs. The first step is to learn node embeddings using a GNN to obtain soft node correspondences between two graphs. The second step is to iteratively refine them using the constraints of matching consensus in local neighborhoods between graphs. The overall refining process resembles the classic graph matching algorithm of graduated assignment (Gold & Rangarajan, 1996), but generalizes it using deep neural representation. Experiments show that the proposed algorithm performs well on real-world tasks of image matching and knowledge graph entity alignment.

The paper is interesting and has some good potential but lacks some important evaluations and analyses. My main concerns are as follows.

1) The consensus in the second stage is crucial?
As the title shows, the main technical contribution lies in the second stage of consensus inducing. But, for the real tasks, in the experiments, the gain by the second stage is not significant or often negligible (L=0 vs. L=10 or 20  in Table 1,2,3). The results of the first stage (L=0) already give better results than all the baselines in many cases, so that most gains appear to come from the usage of GNNs for representation. This makes the major contribution of this work less significant.  I hope the authors justify this. And, I guess that's maybe because the consensus information may also be induced in the first stage by matching nodes with relational presentations learned using a GNN. To see this, the authors may run the second stage only without the first stage.

2) Comparison to the graduated assignment (GA) process
As discussed in 3.3, the proposed neighborhood consensus can be viewed as a generalization of GA of Eq.6 with trainable neural modules. But, it's not actually shown what is the gain by this generalization. This needs to be shown experimentally by substituting the second stage by GA process.

3) Robustness to node addition or removal.
All the experiments look assuming only edges are varied. Is this algorithm robust to node addition or removal, occurring in many practical graph matching problems? This needs to be also discussed.

======================================

The rebuttal succeeds in addressing most of my concerns so that I upgrade my initial rating to weak accept. I hope all the points in the rebuttal are included in the final manuscript.

**Experience Assessment:**

I have published in this field for several years.

**Review Assessment: Checking Correctness Of Derivations And Theory:**

I assessed the sensibility of the derivations and theory.

**Review Assessment: Checking Correctness Of Experiments:**

I carefully checked the experiments.

**Review Assessment: Thoroughness In Paper Reading:**

I read the paper at least twice and used my best judgement in assessing the paper.

---

> ### Author Response · Authors · 2019-11-11
> **Response to Review #1 (Part 1)**
>
> Dear Reviewer,
>
> thanks a lot for reviewing our paper and providing valuable comments. We are working on incorporating your feedback in a new revision of our paper. We would like to provide more explanations to address your concerns.
>
> Main Contribution
> ================
> We emphasize that the main contribution of our work lies indeed in the second stage of our architecture, i.e., refining initial soft correspondences using a trainable message passing scheme that reaches for neighborhood consensus. Our approach allows us to not only distribute local information, but to also distribute global information in the form of node indicator functions/node colorings using purely local operators. The distribution of global information is then used to resolve ambiguities/false matchings made in the first stage of our architecture. In addition, we proposed optimizations to make the consensus stage scale to large real-world instances.
>
> We agree that obtaining initial soft correspondences via similarity scores of node embeddings is not a novel contribution and shouldn't be viewed as one. We will make this more clear in a revised version. We performed additional experiments to emphasize the usefulness of our consensus stage (see below).
>
> Impact of the Consensus Stage
> ===========================
> We argue that our consensus stage has a huge impact on the resulting performance of our model. For example, on the WILLOW-ObjectClass dataset, it at least reduces the error of the initial model ($L=0$) by half across all categories. On the DBP15K dataset, it consistently improves the model's performance by 4 percentage points on average which we claim to be highly significant.
>
> We nonetheless agree with Reviewer #1 that those improvements should be more significant when using weaker baselines. To verify this, we conducted additional experiments as suggested where we replaced the first stage GNN module with a weaker MLP. The results are shown below which we will include in the final manuscript.
>
> Hits@1 on the WILLOW-ObjectClass dataset:
> ----------------------------+------------------+-----------------+------------------+------------------+
> $\Psi_{\theta_1}$ = MLP                  |  Motorbike   |        Car        |        Duck      |   Winebottle  |
> ----------------------------+------------------+-----------------+------------------+------------------+
>                       $L=0$    | 56.85 ± 2.65 | 73.44 ± 2.65 | 71.93 ± 2.10 | 86.10 ± 1.25 |
> ----------------------------+------------------+-----------------+------------------+------------------+
> isotropic      $L=10$  | 80.34 ± 2.34 | 81.31 ± 2.48 | 81.16 ± 2.55 | 93.53 ± 1.38 |
> isotropic      $L=20$  | 82.24 ± 3.06 | 82.49 ± 3.70 | 81.84 ± 2.92 | 95.14 ± 1.58 |
> ----------------------------+------------------+-----------------+------------------+------------------+
> anisotropic  $L=10$  | 87.15 ± 3.27 | 91.56 ± 2.92 | 88.36 ± 2.55 | 96.57 ± 0.83 |
> anisotropic  $L=20$  | 94.16 ± 3.03 | 94.23 ± 2.14 | 90.03 ± 2.21 | 97.24 ± 0.84 |
> ----------------------------+------------------+-----------------+------------------+------------------+
>
> It can be seen that we can still obtain SOTA results even when starting from a weak initial baseline. Here, the consensus stage improves the initial matchings significantly, with nearly up to 40 percentage points improvements on the Motorbike class. However, it is worth noting that good initial matchings do help the consensus stage to improve its performance further, which stresses the importance of our two-stage approach. Furthermore, starting from weak initial matchings takes significantly more refinement steps to converge (as can be seen by the difference between $L=10$ and $L=20$).
>
> It should be noted that the first stage cannot infer any information about consensus. In order to check for consensus, an initial matching is needed that can be tested. The first stage siamese network has no information flow between both heads until the feature matching. The second stage can rerank those hypotheses based on additional information: matching agreement in neighborhoods. It can not be applied without an initial ranking of correspondences.

---

> ### Author Response · Authors · 2019-11-11
> **Response to Review #1 (Part 2)**
>
>
> Comparison to the Graduated Assignment Algorithm
> =============================================
>
> As stated in Section 3.3, our algorithm can be viewed as a generalization of the Graduated Assignment (GA) algorithm extending it by trainable parameters. To evaluate the impact of this we replaced $\Psi_{\theta_2}$ by the fixed function $F(\mathbf{X}, \mathbf{A}, \mathbf{E}) = \mathbf{A} \mathbf{X}$ in the second stage of our approach. Results are shown below:
>
> Hits@1 on the WILLOW-ObjectClass dataset:
> --------------------------------+------------------+-----------------+------------------+------------------+
> Isotropic Methods         |  Motorbike   |        Car        |       Duck        | Winebottle   |
> --------------------------------+------------------+-----------------+------------------+------------------+
>                            $L=0$   | 83.89 ± 2.65 | 84.97 ± 3.00 | 86.80 ± 2.41 | 94.55 ± 1.46 |
> --------------------------------+------------------+-----------------+------------------+------------------+
> Fixed $F$             $L=10$ | 91.33 ± 2.10 | 88.02 ± 2.46 | 89.46 ± 1.69 | 96.58 ± 0.87 |
>                            $L=20$ | 91.51 ± 2.09 | 87.29 ± 4.22 | 89.44 ± 2.43 | 97.04 ± 0.47 |
> --------------------------------+------------------+-----------------+------------------+------------------+
> Trainable $\Psi_{\theta_2}$  $L=10$ | 92.73 ± 2.60 | 93.18 ± 3.01 | 91.80 ± 2.00 | 97.97 ± 0.78 |
>                            $L=20$ | 93.10 ± 2.50 | 93.77 ± 3.18 | 92.11 ± 2.33 | 98.16 ± 0.78 |
> --------------------------------+------------------+-----------------+------------------+------------------+
>
> Hits@1 on the DBP15K dataset:
> --------------------------------+-----------+------------+-----------+-----------+-----------+-----------+
> Method                           | ZH->EN | EN->ZH | JA->EN | EN->JA | FR->EN | EN->FR |
> --------------------------------+-----------+------------+-----------+-----------+-----------+-----------+
>                            $L=0$   | 72.53    | 67.80     | 73.70    | 70.01    | 86.39    | 84.23    |
> --------------------------------+-----------+------------+-----------+-----------+-----------+-----------+
> Fixed $F$            $L=10$ | 72.92    | 68.80     | 74.91    | 71.38    | 86.54    | 84.86    |
> --------------------------------+-----------+------------+-----------+-----------+-----------+-----------+
> Trainable $\Psi_{\theta_2}$  $L=10$ | 77.16    | 71.77     | 77.36    | 73.93    | 89.12    | 87.50    |
> --------------------------------+-----------+------------+-----------+-----------+-----------+-----------+
>
> As one can see, using trainable neural networks $\Psi_{\theta_2}$ consistently improves upon the results of using the fixed-function message passing scheme. We will add those results to the paper.
>
> Using trainable neural networks instead of the GA process has further advantages:
> * In real-world applications it is often difficult to find meaningful similarities between node and edge features. Our approach learns how to make use of (continuous) features and uses them to guide the refinement procedure further.
> * It allows us to choose from a variety of task-dependent GNN operators, e.g., for learning geometric/edge conditioned patterns or for fulfilling injectivity requirements. The theoretical expressivity discussed in Section 5 could even be enhanced by making use of higher-order GNNs, which we leave for future work.

---

> ### Author Response · Authors · 2019-11-11
> **Response to Review #1 (Part 3)**
>
>
> Robustness to Node Addition or Removal
> ===================================
>
> Our algorithm is not only robust to edge variations, but also to the addition or removal of nodes. This can be explained by the fact that unmatched nodes do not have any influence on the neighborhood consensus error since those nodes do not obtain a color from the functional map given by $\mathbf{S}$. Our neural architecture is able to detect and gradually decrease any false positive influence of these nodes in the refinement stage. Please note that the PascalVOC and DBP15K datasets already contain graph-pairs of varying sizes.
>
> We further verified this experimentally on a synthetic toy dataset following a similar experimental setup to [1], where we additionally add $q$% nodes to the target graph and inter-connect those nodes with all other nodes based on the given Erdős–Rényi edge probability $p$.
>
> Hits@1 on synthetic graphs with $|\mathcal{V}_s|=50$, $p=0.2$:
> -------------------------+------------+------------+-------------+------------+-------------+
> Refinement steps | $q=0.1$ | $q=0.2$ | $q=0.3$ | $q=0.4$ | $q=0.5$ |
> -------------------------+------------+------------+-------------+------------+-------------+
> $L=0$                     |  78.97    | 55.46      |  42.04     | 31.14      | 26.10      |
> $L=10$                   | 100.00   | 100.00   |  99.66     | 98.98      | 98.94      |
> -------------------------+------------+------------+-------------+------------+-------------+
>
> Hits@1 on synthetic graphs with $|\mathcal{V}_s|=100$, $p=0.1$:
> -------------------------+------------+------------+-------------+------------+-------------+
> Refinement steps | $q=0.1$ | $q=0.2$ | $q=0.3$ | $q=0.4$ | $q=0.5$ |
> -------------------------+------------+------------+-------------+------------+-------------+
> $L=0$                     | 72.47     | 43.68     | 32.56       | 22.48      | 19.43      |
> $L=10$                   | 100.00   | 100.00   | 99.99      | 99.82      | 99.63      |
> -------------------------+------------+------------+-------------+------------+-------------+
>
> As can be seen, our consensus stage is extremely robust to the addition/removal of nodes while the first stage alone has major difficulties in finding the right matching.
>
> [1] Xu et al.: Gromov-Wasserstein Learning for Graph Matching and Node Embedding (ICML'19)

---

### Official Review · AnonReviewer2 · 2019-10-31
**Official Blind Review #2**

**Rating:** 6

**Review:**

The authors proposed a message passing neural network-based graph matching methods. The overall framework can be viewed as a graph siamese network, where two set of points are passing through the same graph neural network, and then two new embeddings are generated. Using the two embedding the similarity between points can be computed and then the final matching can be generated.

The overall structure of this paper is similar to [1] and [2], the authors should discuss the difference of the proposed with these two papers, if it is possible, the authors may try to compare with these two methods in experiments. Currently, I think the main contribution of the paper should be the new message-passing scheme (in Sec. 3.2). However, from the current experiment, I can not see if the performance improvement is from the new message-passing scheme.

In fact, the message passing scheme is also related to the dual decomposition framework, which is previously used in the graph matching area. For example, in [3], a message-passing algorithm derived from dual decomposition is used to solve the graph matching problem. The authors may also consider add some discussion the difference between message-passing derived from dual decomposition and the message passing in the graph neural network.

==================================== After Revision ==============================================
In the new experiment, the authors proved that the new message-passing scheme (i.e. the consensus stage) in Sec 3.2 can successfully improve the performance by refining original assignment. Thus I modify the score to weak accept.


[1] Wang, Yue, and Justin M. Solomon. "Deep Closest Point: Learning Representations for Point Cloud Registration.", ICCV 2019,
[2] Zhen Zhang, and Wee Sun Lee. "Deep Graphical Feature Learning for the Feature Matching Problem.", ICCV 2019
[3] Paul Swoboda et. al. "A study of Lagrangean decompositions and dual ascent solvers for graph matching.", CVPR 2017

**Experience Assessment:**

I have published in this field for several years.

**Review Assessment: Checking Correctness Of Derivations And Theory:**

I assessed the sensibility of the derivations and theory.

**Review Assessment: Checking Correctness Of Experiments:**

I carefully checked the experiments.

**Review Assessment: Thoroughness In Paper Reading:**

I read the paper thoroughly.

---

> ### Author Response · Authors · 2019-11-11
> **Response to Review #2 (Part 1)**
>
> Dear Reviewer,
>
> thanks a lot for reviewing our paper and providing valuable comments. We are working on incorporating your feedback in a new revision of our paper. We would like to provide more explanations to address your concerns.
>
> Main Contribution
> ================
> We emphasize that the main contribution of our work lies indeed in the second stage of our architecture, i.e., refining initial soft correspondences using a trainable message passing scheme that reaches for neighborhood consensus. Our approach allows us to not only distribute local information, but to also distribute global information in the form of node indicator functions/node colorings using purely local operators. The distribution of global information is then used to resolve ambiguities/false matchings made in the first stage of our architecture. In addition, we proposed optimizations to make the consensus stage scale to large real-world instances.
>
> We agree that obtaining initial soft correspondences via similarity scores of node embeddings is not a novel contribution and shouldn't be viewed as one. We will make this more clear in a revised version.
>
> In addition, we argue that our consensus stage has a huge impact on the resulting performance of our model. For example, on the WILLOW-ObjectClass dataset, it at least reduces the error of the initial model ($L=0$) by half across all categories. On the DBP15K dataset, it consistently improves the model's performance by 4 percentage points on average which we claim to be highly significant. We performed additional experiments as suggested by Reviewer #1 to emphasize the usefulness of our consensus stage (please see the response to Review #1 for more details).
>
> Relation to [1] Wang and Solomon: Deep Closest Point: Learning Representations for Point Cloud Registration (ICCV'19)
> ======================================================================================================
> This work tackles the problem of finding an unknown rigid motion between point clouds by first matching points followed by a differentiable SVD module. We agree that this work tackles the feature matching procedure in a similar fashion as we do in our initial feature matching procedure based on inner product similarity scores. Additionally, this work leverages a Transformer module to let point clouds know about each other before feature matching takes place.
>
> Our work differs in that we introduce a consensus stage to resolve ambiguities in matchings after the initial matching procedure based on neighborhood consensus. Hence, our method could be used to improve the results of [1] further. In order to resolve ambiguities, [1] can only rely on the least squares optimization, inherentely utilizing the rigid embedding of the point cloud in $\mathbb{R}^3$, which does not exist for general graphs and is not required for our approach. In addition, our approach is highly scalable due to only operating an local neighborhoods, while the Transformer module operates on the whole point cloud in a global fashion.
>
> Due to the different task and difference in assumptions, we do refrain from an in-depth experimental comparison. We will nonetheless discuss the similarities/differences to this work further in our related work.
>
> [1] Wang and Solomon: Deep Closest Point: Learning Representations for Point Cloud Registration (ICCV'19)

---

> ### Author Response · Authors · 2019-11-11
> **Response to Review #2 (Part 2)**
>
>
> Relation to [2] Zhang and Lee: Deep Graphical Feature Learning for the Feature Matching Problem (ICCV'19)
> ============================================================================================
> We thank Reviewer #2 for pointing us to this recent work which we were not aware of yet. Here, the authors propose a compositional message passing algorithm that maps point coordinates into a high-dimensional space. The final matching procedure is done by computing the pairwise inner product between point embeddings, proposing enhancements to the initial embedding stage. It does not apply a refinement stage for soft correspondences. Hence, we see improvements in the architecture of the first stage as orthogonal to our work.
>
> To compare our approach, we replicated the experimental setup of [2]. We train our unmodified anisotropic keypoint architecture on the synthetic keypoint training setup from [2] and evaluate the final model on the Pascal PF dataset [4], using only point coordinates as input.
> Overall, our consensus architecture improves upon the state-of-the-art results of [2] on almost all categories while our $L=0$ baseline is weaker than the results reported in [2]. We report the final results below and will include them in the final version.
>
> Hits@1 on the Pascal PF dataset:
> --------------------+---------+--------+----------+--------+-------+----------+--------+-------+-------+---------+--------+
> Method            | mean | aero | bicycle | bird | boat | bottle | bus  | car   | cat   | chair  | cow  |
> --------------------+---------+--------+----------+--------+-------+----------+--------+-------+-------+---------+--------+
> DGFM [2]         | 88.5    | 76.1 |  89.8    | 93.4  | 96.4 |  96.2    | 97.1  | 94.6 | 82.8 |  89.3  | 96.7  |
> --------------------+---------+--------+----------+--------+-------+----------+--------+-------+-------+---------+--------+
>            $L=0$   | 86.7    | 64.6 |  86.9    | 76.6  | 88.5 |  96.0    | 98.4  | 91.4 | 89.6 |  93.4  | 77.2   |
> Ours  $L=10$ | 95.4    | 83.6 |  92.0    | 94.2  | 98.2 |  99.3    | 99.3  | 98.7 | 98.5 |  99.8  | 96.3   |
>            $L=20$ | 95.5    | 83.0 |  92.1    | 92.9  | 98.2 |  99.3    | 99.0  | 98.7 | 99.2 | 100.0 | 96.3   |
> --------------------+---------+--------+----------+--------+-------+----------+--------+-------+-------+---------+--------+
>
> --------------------+---------+---------+--------+---------+-----------+-----------+---------+---------+---------+--------+--------+
> Method            | mean | table | dog  | horse | m-bike | person | plant | sheep | sofa  | train  |  tv    |
> --------------------+---------+---------+--------+---------+-----------+-----------+---------+---------+---------+--------+--------+
> DGFM [2]         | 88.5    | 89.7   | 79.5  | 82.6   |  83.5     |  72.8      | 76.7   | 77.1    |  97.3  | 98.2  | 99.5  |
> --------------------+---------+---------+--------+---------+-----------+-----------+---------+---------+---------+--------+--------+
>            $L=0$   | 86.7    | 97.9   | 85.7  | 73.3   |  76.8     |  69.4      |  97.4   | 76.4   |  85.1  |  91.7  | 97.4  |
> Ours  $L=10$ | 95.4    | 100.0 | 98.5  | 86.8   |  87.0     |  87.8      | 100.0  | 79.4   |  99.6  | 100.0 | 98.9 |
>            $L=20$ | 95.5    | 99.5   | 98.9  | 86.8   |  86.4     |  89.0      | 100.0  | 76.5   | 100.0 | 100.0 | 99.3 |
> --------------------+---------+---------+--------+---------+-----------+-----------+---------+---------+---------+--------+--------+
>
> In addition, it shows that our method works also well even when not taking any visual information into account. Besides, training converges significantly faster in comparison to [2]. Our algorithm does only make use of 64 000 synthetic training examples, while [2] uses 9 million examples and reports convergence not until about 4.5 million examples.
>
> Relation to [3] Swoboda et al.: A Study of Lagrangean Decompositions and Dual Ascent Solvers for Graph Matching (CVPR'17)
> ===========================================================================================================
> Thanks for mentioning the reference. We agree that the message passing algorithms presented in [3] shares some similarities with our message passing architecture on an abstract level. However, the dual ascent algorithms also known as "message passing" used here solves the graph matching problem by using MAP-inference, linear assignment problems and (several small) quadratic assignment problems. Therefore, in this case it is difficult to establish such a clear mathematical relation to our method as we were able to show for the GA method. We will add a discussion of [3] to the related work section on graph matching approaches.
>
> [2] Zhang and Lee: Deep Graphical Feature Learning for the Feature Matching Problem (ICCV'19)
> [3] Swoboda et al.: A Study of Lagrangean Decompositions and Dual Ascent Solvers for Graph Matching (CVPR'17)
> [4] Ham et al.: Proposal Flow (CVPR'16)

---

### Decision · Program_Chairs · 2019-12-19

**Decision:**

Accept (Poster)

**Comment:**

The paper proposed an end-to-end network architecture for graph matching problems, where first a GNN is applied to compute the initial soft correspondence, and then a message passing network is applied to attempt to resolve structural mismatch. The reviewers agree that the second component (message passing) is novel, and after the rebuttal period, additional experiments were provided by the authors to demonstrate the effectiveness of this. Overall this is an interesting network solution for graph-matching, and would be a worthwhile addition to the literature.